# B cells extract antigens at Arp2/3-generated actin foci interspersed with linear filaments

**Sophie I Roper[1], Laabiah Wasim[1], Dessislava Malinova[1,2], Michael Way[3], Susan Cox[4], Pavel Tolar[1,2]***

[1]Immune Receptor Activation Laboratory, The Francis Crick Institute, London, United Kingdom; [2]Division of Immunology and Inflammation, Department of Medicine, Imperial College London, London, United Kingdom; [3]Cellular Signalling and Cytoskeletal Function Laboratory, The Francis Crick Institute, London, United Kingdom; [4]Randall Centre for Cell and Molecular Biophysics, King's College London, London, United Kingdom

**Abstract** Antibody production depends on B cell internalization and presentation of antigens to helper T cells. To acquire antigens displayed by antigen-presenting cells, B cells form immune synapses and extract antigens by the mechanical activity of the acto-myosin cytoskeleton. While cytoskeleton organization driving the initial formation of the B cell synapse has been studied, how the cytoskeleton supports antigen extraction remains poorly understood. Here we show that after initial cell spreading, F-actin in synapses of primary mouse B cells and human B cell lines forms a highly dynamic pattern composed of actin foci interspersed with linear filaments and myosin IIa. The foci are generated by Arp2/3-mediated branched-actin polymerization and stochastically associate with antigen clusters to mediate internalization. However, antigen extraction also requires the activity of formins, which reside near the foci and produce the interspersed filaments. Thus, a cooperation of branched-actin foci supported by linear filaments underlies B cell mechanics during antigen extraction.

**\*For correspondence:**
pavel.tolar@crick.ac.uk

**Competing interests:** The authors declare that no competing interests exist.

## Introduction

Generation of high-affinity, protective antibodies requires B cell uptake of antigens via their antigen-specific B cell receptors (BCRs). After antigen binding, the BCR undergoes endocytosis and delivers the antigen into intracellular processing compartments, where antigenic peptides are loaded onto major histocompatibility II (MHC II) proteins (*Hoogeboom and Tolar, 2016*). Presentation of the peptide-MHC II complexes on the surfaces of B cells solicits T cell help and promotes B cell proliferation and differentiation into germinal center, memory and plasma cells.

The uptake of soluble antigen by the BCR occurs predominantly via clathrin-coated pits (CCPs) (*Stoddart et al., 2005*). In contrast, the uptake of antigens from antigen-presenting immune or stromal cells requires additional forces generated by the acto-myosin cytoskeleton through the formation of B cell immune synapses (*Batista et al., 2001*; *Fleire, 2006*; *Natkanski et al., 2013*). Myosin IIa is thought to be central to generation of these forces, because it is not required for endocytosis of soluble antigens, but promotes force-mediated antigen extraction from presenting cells in vitro and in vivo (*Hoogeboom et al., 2018*).

Nevertheless, the organization of the actin cytoskeleton in B cell synapses that leads to antigen extraction is incompletely understood. Synaptic actin polymerization is mediated by two mechanisms: the Arp2/3 complex producing branched actin networks, and formins producing linear actin filaments (*Blanchoin et al., 2014*). It is thought that these distinct actin nucleation pathways are

activated in the synapse periphery and create an outer band of branched actin and inner ring of contractile acto-myosin filaments, respectively (*Bolger-Munro et al., 2019*; *Hoogeboom and Tolar, 2016*). The constant generation and turnover of the actin results in centripetal flow with central actin clearance. Indeed, in naive B cells interacting with stiff experimental substrates, such as glass or glass-supported planar lipid bilayers, F-actin forms a prominent peripheral ring resembling lamellipodia, and an area with contractile arcs, while the synapse center is clear of actin (*Bolger-Munro et al., 2019*; *Fleire, 2006*; *Freeman et al., 2011*; *Treanor et al., 2011*). However, B cell synapses also contain prominent, but poorly characterized F-actin spots (*Keppler et al., 2015*; *Nowosad et al., 2016*; *Treanor et al., 2010*). Similar spots, termed actin foci, were recently observed in T cell synapses on planar lipid bilayers, where they colocalize with T cell receptor microclusters (*Kumari et al., 2015*). Nevertheless, the relevance of this actin organization for B cell antigen extraction is unclear as antigen uptake from these stiff substrates is mechanically inefficient and antigens are instead enzymatically digested out through the release of lytic enzymes into the synaptic cleft (*Spillane and Tolar, 2017*; *Yuseff et al., 2011*). In contrast, studies using soft substrates, such as plasma membrane sheets (PMSs), which promote mechanical antigen extraction similar to that occurring with live antigen presenting cells, found that actin formed abundant foci throughout the synapse with little central clearance (*Natkanski et al., 2013*). These actin foci were highly dynamic and colocalized with membrane invaginations during antigen uptake, suggesting their involvement in antigen extraction. However, how these actin structures cooperate with other actin filaments and with myosin IIa contractility to produce extraction forces is not clear.

One possibility is that myosin IIa causes contraction of formin-generated actin filaments connecting to, or sweeping through the actin foci to generate inward or lateral forces, respectively. This mechanism would be similar to transport of antigen clusters from the synapse periphery to the center by the contractile act-myosin arcs (*Bolger-Munro et al., 2019*; *Murugesan et al., 2016*). Alternatively, inward forces on the BCR can be generated by actin polymerization within the actin foci, with the help of membrane-bending proteins orienting the membrane for invagination (*Hinze and Boucrot, 2018*; *Kaksonen et al., 2006*).

Here we image the F-actin at the B cell synapse using super-resolution microscopy in 2D and 3D, and using live-cell imaging with high temporal resolution to resolve the structures that are relevant for antigen extraction. We show that the B cell immune synapse contains a highly stochastic and dynamic pattern of actin foci interspersed with actin fibers. The pattern originates from transient spots of Arp2/3 activity generating branched actin in the foci and from closely associated formin activity producing the surrounding linear filaments. The actin foci are more intimately associated with antigen endocytosis, however, manipulation of the actin-polymerization pathways shows that both Arp2/3 and formins are needed for antigen uptake. We propose that these two actin polymerization mechanisms create a functional foci-filament network which translates local bursts of actin polymerization into an inward force for antigen internalization.

## Results

### Extraction of antigen is mediated by actin foci interspersed with linear filaments

To understand the structure and dynamics of the F-actin cytoskeleton in B cell synapses specifically during antigen uptake, we imaged primary B cells from Lifeact-expressing mice (*Riedl et al., 2010*) incubated with PMSs loaded with a surrogate antigen, anti-Igκ F(ab')$_2$. Total internal reflection fluorescence (TIRF) microscopy time-lapses showed that after initial cell spreading, between 5 and 20 min after synapse formation, the Lifeact-labeled F-actin was distributed throughout the synapse in a range of small structures including actin foci and short fibers (*Figure 1A*). 3D imaging verified that during these 20 min, B cells were extracting the antigen as previously described (*Figure 1B*). TIRF microscopy with a 100 ms time resolution showed that the F-actin was highly dynamic and that the foci and fibers were appearing stochastically, and often, merged, split or interconverted between each other (*Video 1*, left panel). To analyze the actin structures, we segmented the actin foci and fibers from the Lifeact images using round and linear filtering elements, respectively (*Figure 1C*, *Figure 1—figure supplement 1A*, *Video 1*, right panel). This automated segmentation was more sensitive than manual foci detection, and the identified foci positions were indistinguishable from

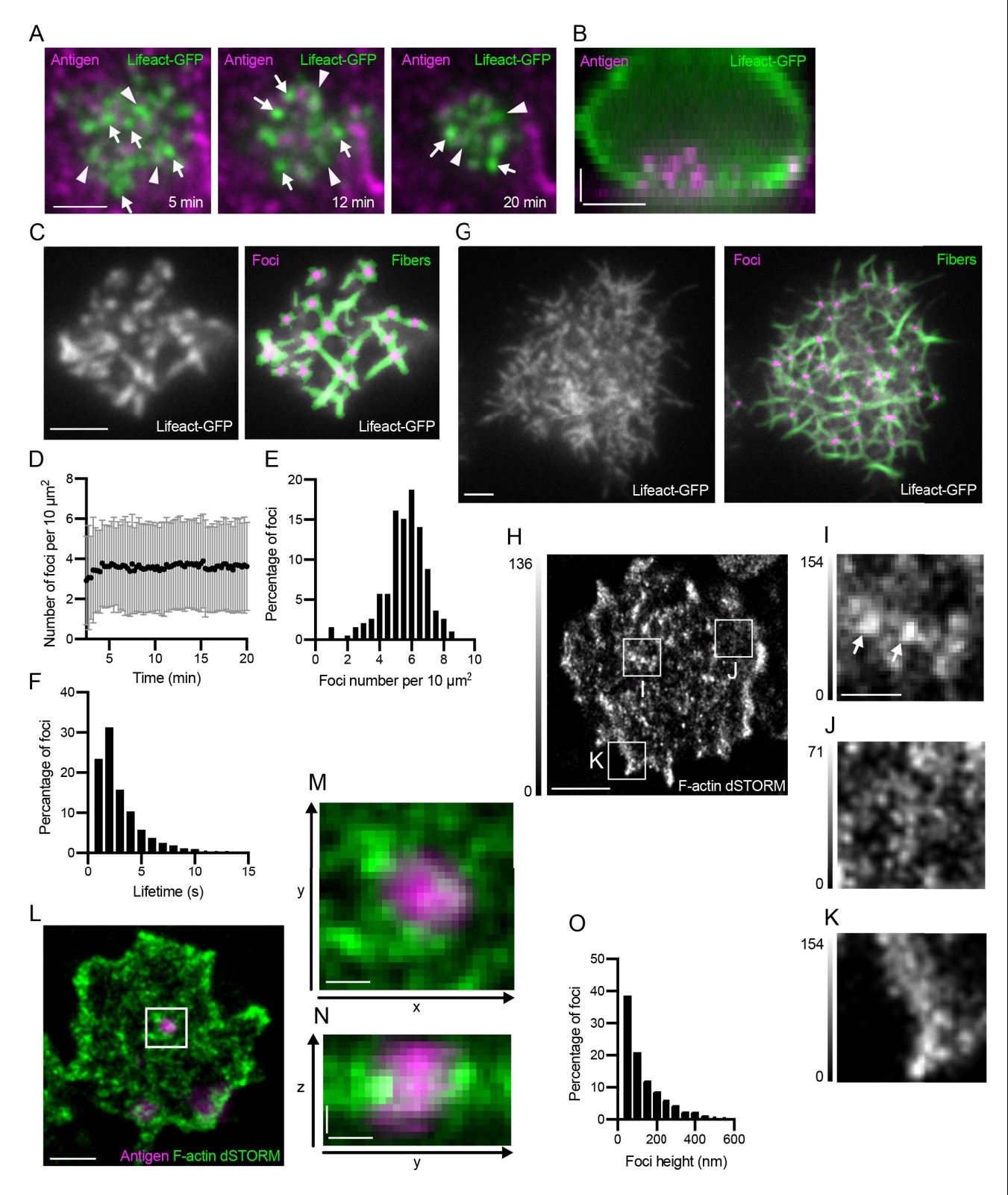

**Figure 1.** Actin structure of the B cell synapse. (**A**) TIRF images of a Lifeact-GFP (green) expressing primary mouse B cell making a synapse with a PMS loaded with the surrogate antigen anti-Igκ F(ab')₂ (magenta), at indicated times from the initial contact. Examples of actin foci and filaments are marked by white arrows and arrowheads, respectively. Scalebar 2 μm. (**B**) Sideview of a Lifeact-GFP (green) expressing B cell after 20 min of contact with a PMS loaded with anti-Igκ F(ab')₂ (magenta). Scalebars 2 μm. (**C**) Example of actin foci and fiber segmentation in primary mouse B cells prepared as in (**A**).
*Figure 1 continued on next page*

*Figure 1 continued*

Left panel shows raw TIRF image of Lifeact-GFP, right panel shows segmented foci (magenta) and fibers (green) overlaid on the original image. Scalebar, 2 µm. (D) Number of actin foci quantified from primary B cells incubated as in (A) at the indicated times after synapse formation. Data points are mean ± SD from 310 cells in one representative experiment. (E) Distribution of the numbers of actin foci in 193 cells from a representative experiment at 20 min after synapse formation. (F) Distribution of the lifetime of actin foci in B cells from the experiment in (E). (G) Actin structure and segmentation of foci and fibers in a Ramos B cell interacting with a PMS loaded with anti-IgM. Scalebar, 2 µm. (H) dSTORM reconstruction of F-actin stained with phalloidin-AlexaFluor647 at the focal plane of a synapse using TIRF illumination of a primary mouse B cell interacting with a PMS loaded with anti-Igκ F(ab')$_2$. Scale bar, 2 µm. (I, J, K) Magnified areas show actin foci (arrows), filament meshwork, and lamellipodia, respectively. Gray intensity scales are different from H and are shown on the left of each image. Scale bar, 0.5 µm. (L) 3D dSTORM reconstruction of F-actin, stained with phalloidin-AlexaFluor647 (green) overlaid onto epifluorescence images of PMS loaded with anti-Igκ F(ab')$_2$ (antigen, magenta). Image shows a section containing actin 250–300 nm above the PMS and the corresponding epifluorescence section. Scale bar 1 µm. (M) Magnified area containing an antigen cluster lifted from the PMS, surrounded by actin foci. (N) Vertical section through this area shown as maximum intensity projection along x. Scale bars in (M), (N) 0.2 µm. (O). Histogram of foci size in the z-direction (height). Data are from 1369 foci in 58 cells from one representative experiment.

The online version of this article includes the following figure supplement(s) for figure 1:

**Figure supplement 1.** Actin foci and fiber segmentation.

---

segmentation results produced by four individual researchers (*Figure 1—figure supplement 1B*). We therefore used the automated segmentation as an unbiased quantification tool to analyze all imaged cells. Segmentation of actin foci and fibers from time-lapse microscopy of the primary mouse B cells showed that foci numbers per area remained stable at about 5 per 10 µm2 during the 20 min of imaging (*Figure 1D,E*), but that most individual foci persisted for less than 5 s (*Figure 1F*). A similar pattern of foci and interconnecting fibers was also observed in the human Ramos cell line, a commonly used B cell lymphoma line, although these cells were larger and had more filopodia emanating from the sides of the synapse (*Figure 1G*).

To generate a high-resolution view of these F-actin structures, we used the super-resolution localization technique direct stochastic optical reconstruction microscopy (dSTORM) using phalloidin-AlexaFluor647 staining of fixed cells (*Figure 1H*). Super-resolution reconstruction of the F-actin showed numerous dense actin foci (*Figure 1I*), interspersed by a sparse network of fibers (*Figure 1J*). This pattern filled the body of the synapse and was surrounded by thin lamellipodia (*Figure 1K*).

Dual color live-cell TIRF microscopy indicated that although the actin structures did not overtly colocalise with antigen, some actin foci did transiently coincide with antigen clusters (*Video 2*), in line with previous data. To understand the structure of the F-actin at the sites of antigen extraction in more detail, we used 3D super-resolution imaging of F-actin and overlaid the super resolution actin image with epifluorescence stacks of antigen images (*Figure 1L*, *Video 3*). Antigen clusters that were being internalized could be identified, because they were lifted several hundred nanometers from the PMSs. Such lifted clusters of antigen were surrounded by dense F-actin foci from the sides (*Figure 1M,N*). Measurement of the height of all the actin foci showed that more than 20% of them extended more than 200 nm into the cell (*Figure 1O*). This suggests that foci are the actin structures associated with antigen movement into the cell and thus likely correspond to actin spots previously associated with membrane invaginations (*Natkanski et al., 2013*). The 3D STORM reconstruction also indicates that the actin foci interact with antigen clusters from the sides or from below, suggesting that they are involved in the transport of the antigen to the interior of the cell.

## Arp2/3 and formins generate the distinct actin structures of the B cell synapse

To determine the mechanisms by which the actin foci and fibers are generated, we blocked actin polymerization in PMS-attached B cells either with CK666, which specifically inhibits the activity

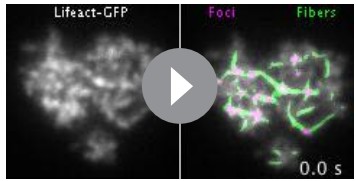

**Video 1.** Actin dynamics at the B cell synapse. Left panel shows raw TIRF image stream with 100 ms time resolution of a primary mouse B cell expressing Lifeact-GFP spread on anti-Igκ F(ab')$_2$. Right panel shows the cell image overlaid with segmented actin foci and fibers.

https://elifesciences.org/articles/48093#video1

---

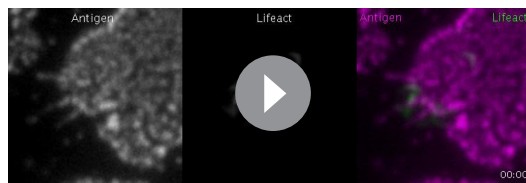

**Video 2.** Actin foci transiently colocalise with antigen clusters. Video shows TIRF timelapse of lifeact-GFP (lifeact) in a primary mouse B cell interacting with anti-Igκ F(ab')₂-AlexaFluor647-loaded PMS (antigen). Elapsed time is shown in minutes and seconds. Circles highlight actin foci colocalizing with an antigen cluster.
https://elifesciences.org/articles/48093#video2

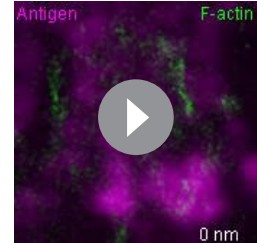

**Video 3.** 3D super-resolution image of actin (green) overlaid with epifluorescence images of antigen (magenta). The video steps through 50 nm-thick sections of the cell shown in *Figure 1L*.
https://elifesciences.org/articles/48093#video3

of the Arp2/3 complex, or with SMIFH2, which inhibits formins. Super-resolution imaging showed that CK666 treatment resulted in a loss of actin foci, which were instead replaced by abundant actin fibers (*Figure 2A*). In contrast, treatment with SMIFH2 primarily dispersed the fibers in between foci (*Figure 2A*). Quantification of fast timelapse imaging of CK666-treated B cells confirmed reduction in foci numbers (*Figure 2B*) and showed that the residual fibers were dynamic, growing, shrinking and curving rapidly (*Video 4*). On the other hand, in SMIFH2-treated B cells, foci numbers were increased, although their lateral movement was arrested and their dynamics stunted (*Video 4*, *Figure 2B*). These results suggested that the actin foci are branched-actin structures generated by the Arp2/3 complex, while the interspersed fibers are linear actin filaments or filament bundles produced by formins.

Similar to primary mouse B cells, treatment of Ramos cells with CK666 reduced foci numbers (*Figure 2—figure supplement 1A,B*, *Video 5*). Treatment with SMIFH2 led to a small decrease, rather than increase in the number of foci, possibly because in these cells the foci became elongated and motile (*Figure 2—figure supplement 1A,B*, *Video 5*), consistent with the idea that formins regulate foci dynamics.

To confirm the inhibitor treatment, we targeted the Arp2/3 complex and formins genetically. The Arp2/3 complex was acutely targeted by CRISPR in Ramos cells expressing Cas9 using three different guide RNAs (gRNAs) specific for *ARPC2*, an essential component of the complex. All three gRNAs greatly reduced the average levels of the ARPC2 protein in the targeted cell population (*Figure 2C*). All three gRNAs reduced the numbers of actin foci and promoted production of actin fibers across the synapse and in filopodia (*Figure 2D,E*). Thus, ARPC2 is essential for production of actin foci in the B cell synapse.

Formins are a family with multiple members that are in many aspects functionally redundant. We targeted two formins highly expressed in B cells including Ramos cells (*Brazão et al., 2016*; *Klijn et al., 2015*), DIAPH1 and FMNL1. *DIAPH1* was successfully targeted in Ramos cells with one gRNA and *FMNL1* with two gRNAs (*Figure 2C*). We also generated Ramos cells lacking both DIAPH1 and FMNL1 by re-targeting the DIAPH1-targeted cells with two different *FMNL1* gRNAs. Imaging F-actin and quantification of actin foci revealed that targeting of the formins resulted in little change of the synaptic actin pattern (*Figure 2F*), although quantification showed a subtle decrease in the number of actin foci in cells targeted with the DIAPH1 gRNA, and a small increase in cells targeted with FMNL1 or both DIAPH1 and FMNL1 gRNAs (*Figure 2G*). Therefore, neither DIAPH1 nor FMNL1 are required for the formation of actin foci, and they are redundant in production of the filaments outside of the foci.

## Dynamics of Arp2/3 and formins account for the actin architecture of the B cell synapse

To observe the role of Arp2/3 and formins in actin dynamics directly, we transduced Ramos cells with constructs of ARPC2-mRuby or DIAPH1-mRuby and analyzed their localization in phalloidin-stained cells interacting with anti-IgM loaded PMSs. ARCP2-mRuby localized predominantly in round or slightly elongated spots that corresponded to phalloidin-labeled actin foci (*Figure 3A*). ARPC2-mRuby also closely followed the dynamics of actin in foci visualized in time-lapse imaging of Ramos

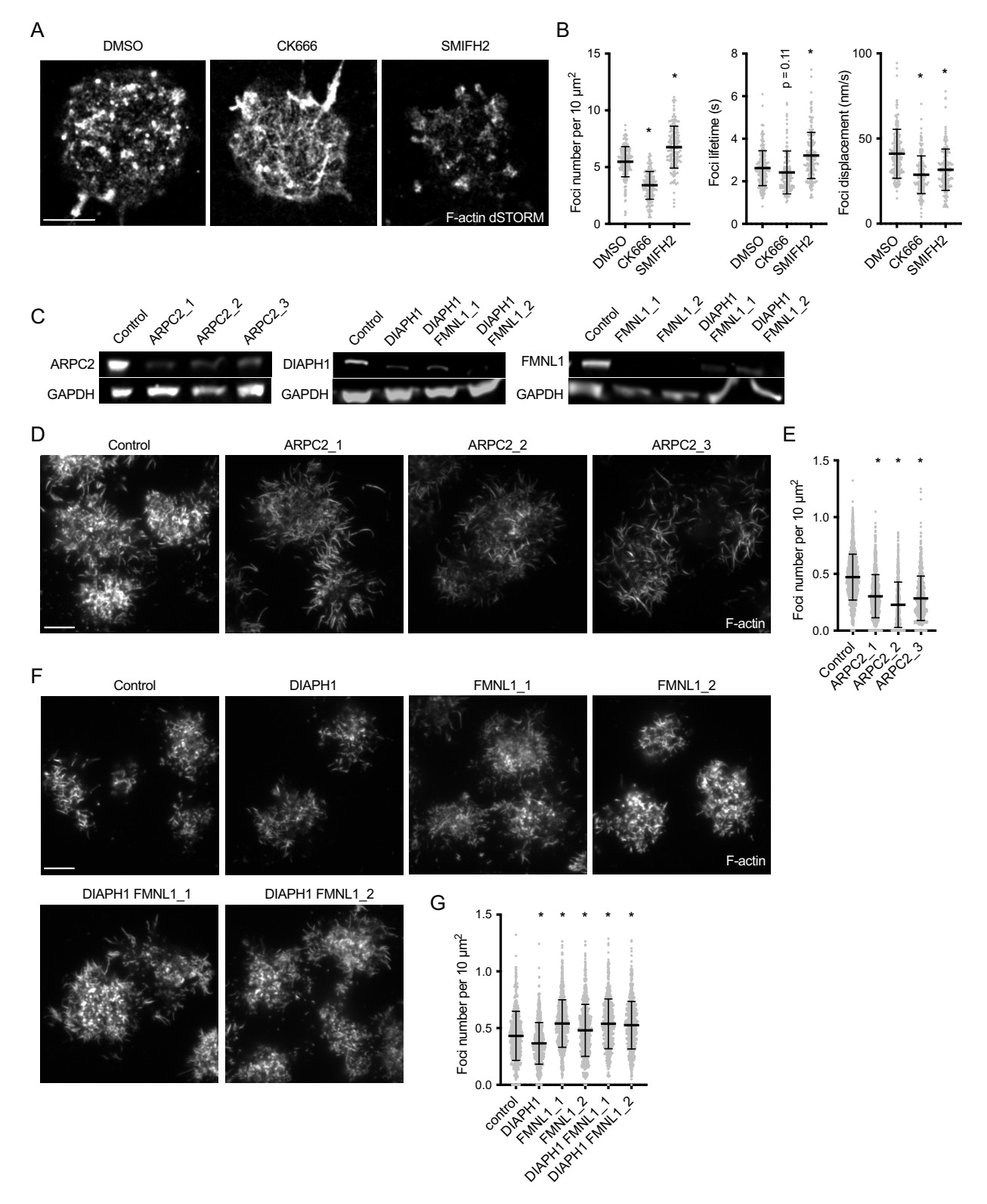

**Figure 2.** Arp2/3 is responsible for actin foci generation, while formins produce the interspersed actin filaments. (**A**) dSTORM reconstruction of F-actin stained with phalloidin-AlexaFluor647 in synapses of primary mouse B cells interacting for 20 min with PMS loaded with anti-Igκ F(ab')₂. The cells were treated with the indicated inhibitors from 5 min after initial spreading. Scale bar, 2 μm. (**B**) Quantification of foci characteristics obtained from TIRF timelapses of live cells treated with inhibitors as in (**A**). Plots show foci numbers, mean lifetimes, and mean displacements per cell. (**C**) Immunoblot of

*Figure 2 continued on next page*

*Figure 2 continued*

lysates of Cas9 Ramos cells transfected with the indicated gRNAs, developed with anti-ARPC2 and anti-GAPDH antibodies. (**D, F**) TIRF images of F-actin stained with phalloidin-AlexaFluor647 in synapses of a Cas9-expressing Ramos B cells transduced with the indicated gRNAs. Cells were imaged after interacting for 20 min with PMS loaded with anti-IgM F(ab')$_2$. Scale bars, 5 µm. (**E, G**) Quantification of Ramos actin foci numbers per cell area. Data in B, E, G show values from individual cells pooled from two (**B, G**) or three (**E**) experiments. Bars indicate means and SDs. P, significance in one-way ANOVA compared to DMSO or control, *, p<0.0001.

The online version of this article includes the following figure supplement(s) for figure 2:

**Figure supplement 1.** Actin architecture of Ramos B cell synapses after Arp2/3 and formin inhibition.

cells co-expressing Lifeact-GFP (*Figure 3B*, *Video 6*). The ARPC2-mRuby-positive actin foci were surrounded by short, ARPC2-mRuby-negative actin fibers, which were frequently seen dynamically emanating from the foci and sometimes transiently connecting to other foci (*Figure 3C*). Simultaneous labeling of the Ramos B cell plasma membrane using the lipid dye DiD indicated that while in the cell periphery the fibers grew into filopodia, in the center of the synapse, the short fibers did not correspond to membrane structures (*Figure 3—figure supplement 1*).

DIAPH1-mRuby localization was more diffuse than that of ARPC2, but some DIAPH1-mRuby spots were also visible and they often overlapped with actin foci (*Figure 3D*). However, live cell imaging indicated that DIAPH1 spots followed the dynamics of actin foci less precisely than ARPC2 spots did (*Figure 3E*, *Video 7*). The spots of DIAPH1-mRuby were also seen producing actin fibers growing out of them, including filopodia (*Figure 3F*, *Video 7*).

Quantification confirmed that ARPC2-mRuby was significantly enriched in actin foci, while it was depleted in fibers (*Figure 3G*). In addition, dual segmentation of the ARPC2 spots and actin foci showed that a significant fraction of actin foci directly colocalized with ARPC2 spots and vice versa (*Figure 3H*). DIAPH1-mRuby was also enriched in actin foci as compared to fibers (*Figure 3G*) and DIAPH1-mRuby spots significantly colocalized with actin foci (*Figure 3I*). However, its enrichment in foci and rates of colocalization were significantly lower than that of ARPC2.

Thus, actin foci contain predominantly branched actin marked with ARPC2, and they represent the dominant branched-actin structures of the B cell synapse. Nevertheless, the foci also associate with DIAPH1, which does not contribute to foci generation, but produces short linear filaments that surround and interconnect them.

## Myosin IIa activity is not required for foci formation

To understand the role of myosin IIa contractility in regulation of the actin architecture and particularly the actin foci, we analyzed the relationship of F-actin to activated myosin IIa using antibodies specific for the phosphorylated myosin II light chain. This imaging showed that most of the active myosin IIa did not overlap with the most prominent F-actin structures, although it was localized in close proximity to both actin foci and actin fibers (*Figure 3—figure supplement 2A,B*). In addition, the inhibition of the ATPase activity of myosin IIa with para-nitroblebbistatin only modestly reduced foci numbers (*Figure 3—figure supplement 2C,D*, *Video 8*), despite strongly impairing antigen extraction (*Figure 3—figure supplement 2E*). Thus, actin foci are not myosin IIa-dependent contractile structures, although they are subtly regulated by its motor activity, which is consistent with their branched actin structure and a rapid turnover.

## BCR signaling regulates actin foci dynamics

To investigate the relationship of the actin architecture to BCR signaling triggered by antigen binding, we stimulated Ramos cells with anti-IgM-coated PMSs and analyzed localization of total tyrosine-phosphorylated proteins as well as phosphorylated BLNK and CD19, two critical adaptors that propagate signaling to the

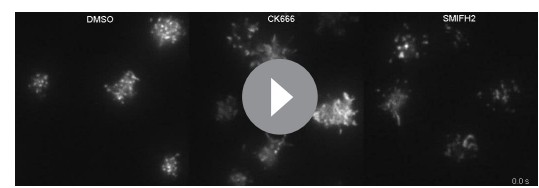

**Video 4.** Actin dynamics after Arp2/3 and formin inhibition. The panels show TIRF timelapses with 100 ms time resolution of Lifeact-GFP in primary mouse B cells spread on anti-Igκ F(ab')$_2$-loaded PMSs and treated with DMSO, CK666 and SMIFH2.
https://elifesciences.org/articles/48093#video4

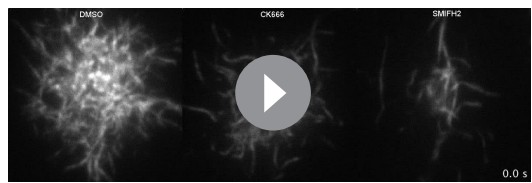

**Video 5.** Actin dynamics in Ramos cells after Arp2/3 and formin inhibition. The panels show TIRF timelapses with sub-second time resolution of Lifeact-GFP in Ramos cells spread on anti-IgM F(ab')$_2$-loaded PMSs and treated with DMSO, CK666 and SMIFH2. Related to **Figure 2—figure supplement 1A**.
https://elifesciences.org/articles/48093#video5

cytoskeleton and downstream activation pathways (**Figure 4A,B**). All phosphotyrosine signals had low colocalization with actin, however, we detected significantly higher enrichment of total phosphotyrosine and phospho-CD19 signals in actin foci than in actin fibers (**Figure 4B**). Phosphorylated BLNK was only weakly enriched in foci, but was still highly excluded from fibers (**Figure 4B**). Thus, actin foci are the predominant actin structure associated with BCR signaling.

However, we observed that the actin architecture composed of foci and interspersed fibers did not require antigen-induced stimulation of the BCR, as it was similar in B cells interacting with PMSs loaded with non-stimulatory anti-MHC I or anti-MHC II antibodies (**Figure 4—figure supplement 1A**, **Video 9**), with numbers of foci even slightly increased (**Figure 4—figure supplement 1B**). The actin foci, were also accentuated by treatment of antigen-activated B cells with the Src-family inhibitor PP2, despite the reduction in total F-actin (**Figure 4—figure supplement 1C,D**, **Video 10**). Inhibition of Syk using the inhibitor P505-15 led to a more modest increase in foci numbers (**Figure 4—figure supplement 1C,D**, **Video 10**). However, the foci in both PP2 and P505-15 treated B cells had increased lifetimes and reduced lateral mobilities (**Figure 4—figure supplement 1E,F**). Thus, while antigen stimulation is not required for foci formation, BCR signaling regulates synaptic actin architecture including foci dynamics.

## Actin foci dynamically colocalize with CCPs
As mentioned above, actin foci dynamically interacted with antigen clusters during extraction, and had a close relationship with BCR signaling, suggesting that ARPC2 is proximal to antigen uptake. We observed that ARPC2 clusters, colocalized with antigen clusters at the time of extraction and endocytosis (**Video 11**), while DIAPH1 did not (**Video 12**). To understand the relationship of the actin foci to CCPs, the major known endocytic mechanism for antigen uptake, we simultaneously imaged F-actin and the clathrin light chain A (CLTA), tagged with mCherry in Ramos cells. Although most actin foci and CCPs did not colocalize with each other (**Figure 4C**), some CCPs were located in the vicinity of actin structures. Quantification showed that CLTA-mCherry signal was significantly enriched in actin foci as compared to actin fibers (**Figure 4D**). Although only about 3% of foci directly colocalized with CCPs, and 7% of CCPs colocalized with actin foci, these rates were significantly above those expected from random co-localization of these structures (**Figure 4E**). Thus, while most actin foci are independent of CCPs, a small number of them coincides with a fraction of CCPs at any given time, consistent with the transient nature of actin polymerization during clathrin-mediated endocytosis.

## Actin foci and fibers jointly regulate antigen uptake
Overall, our data suggested that actin polymerization in foci is important for antigen extraction. We found that internalization of soluble surrogate antigen anti-Igκ F(ab')$_2$ or soluble transferrin by naive mouse B cells was only very modestly sensitive to inhibition of either of the Arp2/3 complex or formins, or to complete block in actin polymerization by cytochalasin D (**Figure 5A**, **Figure 5—figure supplement 1**). Similarly, internalization of soluble anti-IgM F(ab')$_2$ antibodies by Ramos cells was not significantly inhibited by blocking Arp2/3 or formin activity, although some inhibition was achieved by treatment with cytochalasin D (**Figure 5B**). Thus, the uptake of soluble antigen is relatively independent of actin polymerization. In contrast, extraction of the surrogate antigens by either primary cells or Ramos cells from PMSs was completely abolished by blocking actin polymerization with latrunculin A, indicating importance of actin polymerization for force-mediated uptake of membrane antigens (**Figure 5C,D**). The uptake of antigens from PMSs was also markedly sensitive to inhibition of Arp2/3 by CK666 (**Figure 5C,D**). In addition, CRISPR-mediated targeting of ARPC2 in Ramos cells decreased antigen uptake from PMSs (**Figure 5E**), significantly for two of the three

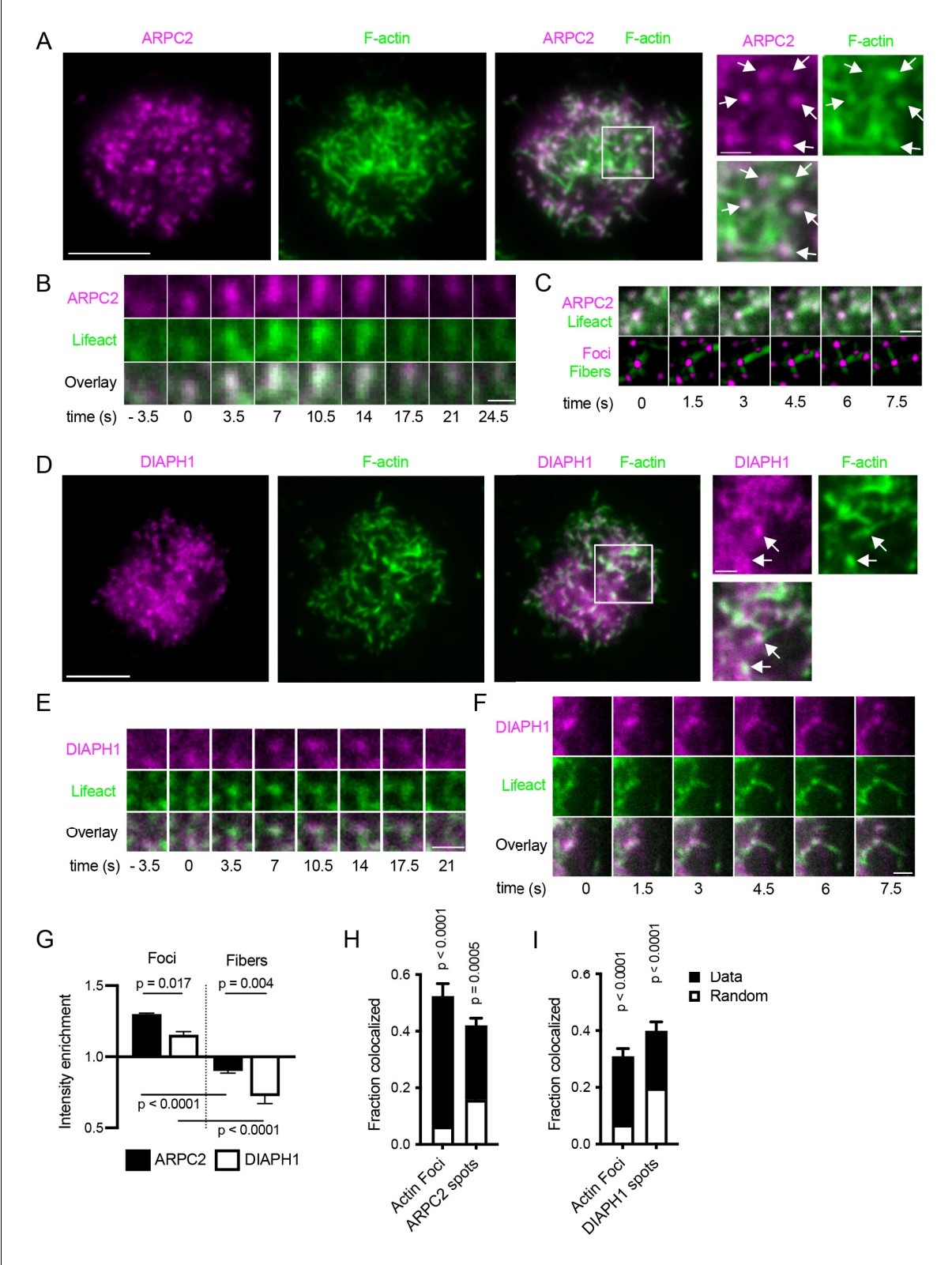

**Figure 3.** Localization and dynamics of ARPC2 and DIAPH1 in synapses of Ramos cells. (**A**) Ramos cells expressing ARPC2-mRuby (magenta) were imaged by TIRF microscopy on PMSs loaded with anti-IgM F(ab')₂. F-actin was stained with phalloidin-AlexaFluor647 (green). Scale bar, 5 μm. Panels on the right show magnified area in the white box. Arrows show ARPC2 clusters colocalized with actin foci. Scale bar 1 μm. (**B**) Example of dynamics of ARPC2-mRuby in a single actin focus visualized with Lifeact-GFP. Time zero corresponds to initial focus formation. Scalebar 1 μm. (**C**) Example of a

*Figure 3 continued on next page*

*Figure 3 continued*
dynamic filament growth from ARPC2-positive actin foci in Ramos cells co-expressing ARPC2-mRuby and Lifeact-GFP. Bottom panel shows results of actin and fiber segmentation. Scalebar 1 μm. (D) Ramos cells expressing DIAPH1-mRuby (magenta) were imaged as in (A). Scale bar, 5 μm. Panels on the right show magnified area in white box. Arrows show clusters of DIAPH1 colocalized with actin foci. Scale bars 1 μm. (E) Example of dynamics of DIAPH1-mRuby in a single actin focus visualized with Lifeact-GFP. Time zero corresponds to initial focus formation. Scalebar 1 μm. (F) Example of a fiber outgrow from a DIAPH1 cluster in *Video 7*. Scalebar 1 μm. (G) Quantification of relative enrichment or depletion of ARPC2-mRuby and DIAPH1-mRuby fluorescence in actin foci and filaments. Data are mean ± SEM from n = 4 experiments each containing 12–213 cells. P, significance from one-way ANOVA. (H, I) Fraction of actin foci colocalized with ARPC2 (G) or DIAPH1 (H) spots and vice versa. Total bar heights show rates of colocalization, white bars indicate colocalization obtained after randomization of locations of the corresponding structures. Values are mean ± SEM of n = 4 of the same experiments as in F. P, significance in two-way repeated measures ANOVA comparing the measured with the randomized data.
The online version of this article includes the following figure supplement(s) for figure 3:

**Figure supplement 1.** Actin fibers within the B cell synapse are not associated with membrane structures.
**Figure supplement 2.** Myosin IIa resides outside of actin foci and its activity is not required for foci formation.

gRNAs. Thus, Arp2/3 activity is critical for uptake of antigens from PMSs. Targeting of DIAPH1 and FMNL1 yielded more variable results with a trend towards inhibition that did not reach statistical significance (*Figure 5F*), consistent with the redundancy of these two formins with other formin family members in regulation of actin polymerization. However, inhibition of formins by SMIFH2 strongly reduced synaptic antigen internalization in both primary B cells and Ramos cells (*Figure 5C,D*).Thus, these results overall support the conclusion that both Arp2/3 and formins are important for the uptake of antigens from the synapse, indicating the importance of both the formation and the dynamics of the actin foci and further support from actin fibers in extraction of membrane-presented antigens.

## Both actin foci and fibers contribute to mechanical activity

To understand how inhibition of Arp2/3 or formins compromises antigen extraction, we analyzed the mechanical activity of the B cells in the immune synapses. Pulling on the antigen during extraction can be visualized by imaging the deformation of the antigen-coated PMSs using the lipid dye DiI (*Natkanski et al., 2013*). Ramos cells expressing Lifeact-GFP produced PMS deformations that appeared as spots, which we previously showed are short, vertical membrane invaginations (*Natkanski et al., 2013*), and also laterally elongated structures, ridges or tubules (*Figure 6A*, *Figure 6—video 1*). The shape of the pulled structures corresponded to the shape of the actin structures associated with them. Spots typically coincided with actin foci (*Figure 6B*, at 15 and 30 s), while shorter tubules associated with elongating foci (*Figure 6B*, at 45 s) and longer tubules or ridges with longer actin fibers (*Figure 6C*). We quantified the rates and location of the PMS deformations by tracking the appearance and movement of these membrane structures with respect to the actin cytoskeleton. Approximately 40% of these pulling events could be assigned to actin foci or fibers using this analysis (*Figure 6D*). About 30% occurred with actin fibers, while less than 10% associated with foci (*Figure 6D*). Most mechanical activity thus comes from areas containing actin fibers. However, correcting for the much smaller area of the cells occupied by foci showed that the actin foci were three times more efficient at applying forces on the PMSs than fibers (*Figure 6E*). Thus, both foci and fibers apply pulling forces on the antigen-presenting membrane, but with different geometry and efficiency.

To understand the role of Arp2/3 and formins in pulling on the antigen-presenting membrane, we analyzed PMS deformations induced by the Ramos cells in the presence of CK666 or SMIFH2. We found that the overall number of membrane pulling events was not affected by either of the inhibitors compared to the DMSO control (*Figure 6F*). However, Ramos cells treated with CK666 often generated elongated

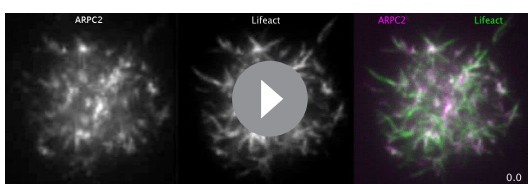

**Video 6.** Localization of ARPC2 in Ramos B cell synapse. TIRF timelapse of Ramos cells co-expressing ARPC2-mRuby and Lifeact-GFP spread on PMSs loaded with anti-IgM F(ab')2. Panels show ARPC2-mRuby (left), Lifeact-GFP (middle), and their overlay (right). Elapsed time is in seconds.
https://elifesciences.org/articles/48093#video6

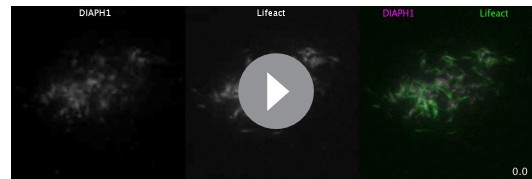

**Video 7.** Localization of DIAPH1 in Ramos B cell synapse. TIRF timelapse of Ramos cells co-expressing DIAPH1-mRuby and Lifeact-GFP spread on PMSs loaded with anti-IgM F(ab')2. Panels show DIAPH1-mRuby (left), Lifeact-GFP (middle), and their overlay (right). Elapsed time is in seconds.
https://elifesciences.org/articles/48093#video7

**Video 8.** Actin dynamics after myosin IIa inhibition. The panels show TIRF timelapses with 100 ms time resolution of Lifeact-GFP in primary mouse B cells spread on anti-Igκ F(ab')2-loaded PMSs and treated with DMSO (left) or pn-Blebbistatin (right).
https://elifesciences.org/articles/48093#video8

membrane structures in the synapse, with some cells producing excessively long and convoluted membrane tubules and ridges (*Figure 6G*, *Figure 6—video 2–4*). To quantify this effect, we classified the DiI images of the PMS in immune synapses as containing only pulled spots, a mixture of spots and short tubules, and a predominance of

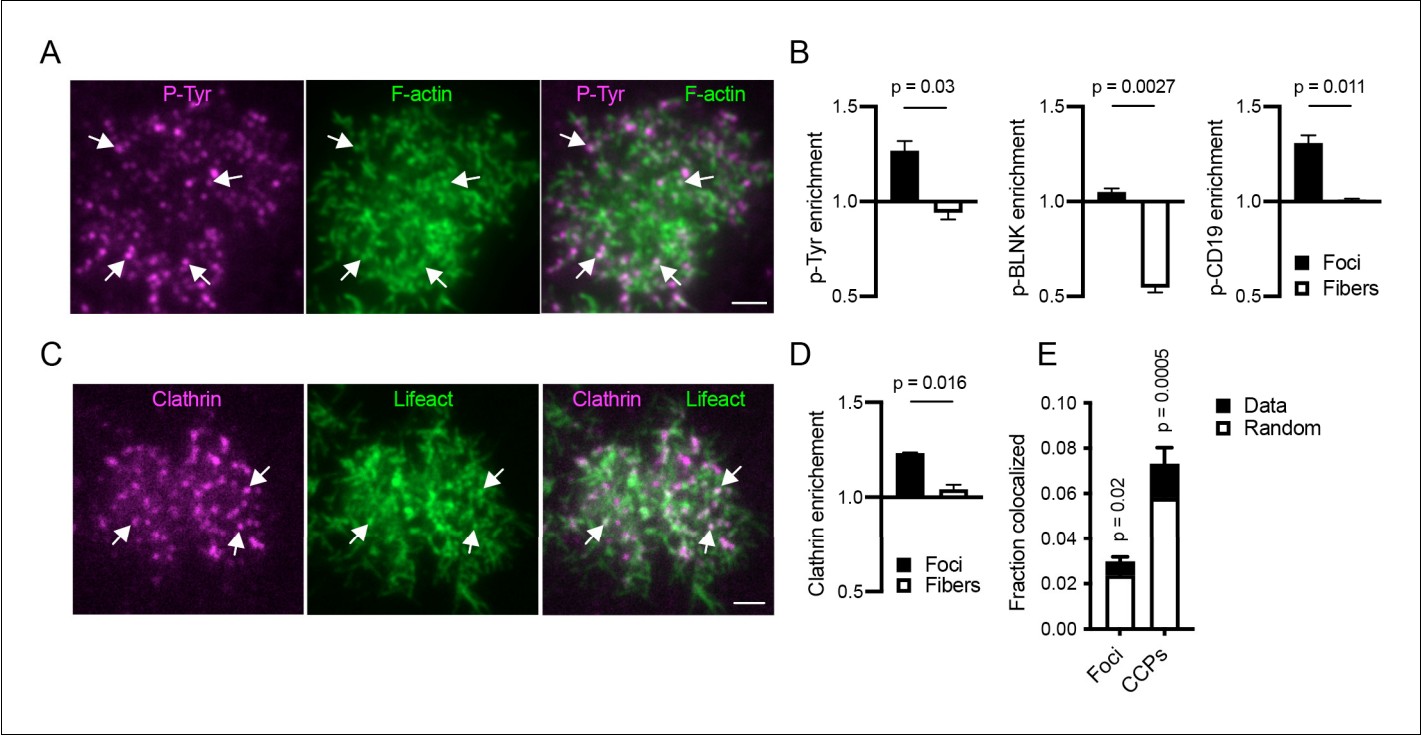

**Figure 4.** Relationship of actin structures to BCR signaling and endocytic complexes. (**A**) TIRF image of a Ramos cell interacting with anti-IgM F(ab')2-loaded PMS and stained for F-actin (green) and phosphotyrosine (magenta). Arrows show colocalization of phosphotyrosine signal with actin foci. Scale bar, 2 μm. (**B**) Quantification of enrichment of the indicated stains in actin foci or fibers. Data are mean ± SEM from n = 4 (phosphotyrosine) or n = 3 (phospho-BLNK and phospho-CD19) experiments. P, significance in paired t-tests. (**C**) TIRF image of a Ramos cell interacting with anti-IgM F(ab')2-loaded PMS and expressing Lifeact-GFP (green) and CLTA-mCherry (magenta). Arrows show colocalization of CCPs with actin foci. Scale bar, 2 μm. (**D**) Quantification of enrichment of clathrin in actin foci and fibers. Mean ± SEM from n = 4 experiments. P, significance in paired t-test. (**E**) Fraction of actin foci colocalized with CCPs and vice versa. Total bar heights show rates of colocalization, the white bars colocalization obtained after randomization of locations of the corresponding structures. Values are mean ± SEM of n = 4 of the same experiments as in D. P, significance in two-way repeated measures ANOVA comparing the measured with the randomized data.

The online version of this article includes the following figure supplement(s) for figure 4:

**Figure supplement 1.** Regulation of actin architecture in primary mouse B cells by BCR signaling.

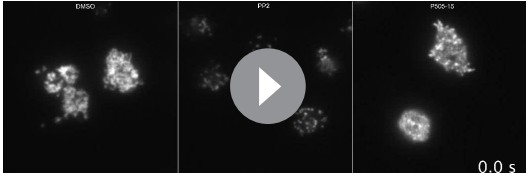

**Video 9.** Actin dynamics in B cells interacting with non-stimulatory substrates. The panels show TIRF timelapses with 100 ms time resolution of Lifeact-GFP in primary mouse B cells spread on anti-Igκ F(ab')2 (left), anti-MHC I (middle), or anti-MHC II (right)-loaded PMSs.
https://elifesciences.org/articles/48093#video9

**Video 10.** Actin dynamics after Src-family and Syk kinase inhibition. The panels show TIRF timelapses with 100 ms time resolution of Lifeact-GFP in primary mouse B cells spread on anti-Igκ F(ab')2-loaded PMSs and treated with DMSO (left), PP2 (middle), or P505-15 (right).
https://elifesciences.org/articles/48093#video10

tubules or ridges (*Figure 6G*). The analysis showed that the pulling activity of CK666-treated Ramos cells resulted in fewer synapses containing only spots and more synapses dominated by tubules (*Figure 6H*). These data are consistent with formin-generated fibers predominantly contributing to long-range antigen movement along the plasma membrane. Overall, while actin polymerization either through Arp2/3 or formins exerts mechanical forces in the immune synapse, coordinated function of both of these actin polymerization mechanisms is required to extract antigen in a manner that leads to efficient endocytosis.

## Discussion

Our data show that after initial cell spreading, the B cell synapse contains a stochastic pattern of foci and fibers. These structures are generated locally in the synapse by transient activities of the Arp2/3 complex and formins, respectively and are important for antigen extraction. Despite their different mechanism of production, the foci and fibers are closely related as they are often generated from each other during their constant turnover. This is also illustrated by colocalization of both Arp2/3 and formins with the actin foci, leading to production of both branched actin and linear filaments from the same sites. This actin behavior produces network-like connectivity that is distinct from the actin structure of CD4 (*Kumari et al., 2015*; *Murugesan et al., 2016*) and CD8 (*Ritter et al., 2015*) T cell synapses and also differs from B cell synapses formed with stiff substrates, where peripheral lamellipodial actin, and acto-myosin contractile rings occupy distinct domains of the synapse (*Bolger-Munro et al., 2019*; *Fleire, 2006*).

We speculate that when B cells extract antigens from live antigen-presenting cells, actin foci and fibers form nodes and edges in a loosely connected network, whose function is required for efficient antigen extraction. In this model, stochastic Arp2/3 activation produces actin foci by local bursts of branching of preexisting short filaments. Branched actin polymerization may either push on the membrane to generate protrusions, similarly as it happens in podosomes (*van den Dries et al., 2019*), or synergize with BCR-dependent signaling and endocytic proteins to produce an invagination of the plasma membrane and extract the antigen. Formin-generated filaments may stabilize the foci by surrounding and connecting them, which, along with crosslinking and tensioning by myosin IIa, could help to direct the foci-generated forces perpendicularly to the plasma membrane. In addition, the linear filament polymerization out of the foci may help to extend the range of the inward movement or enhance the extraction force directly.

This model is supported by our observation and manipulation of the two major regulators of actin polymerization, the Arp2/3 complex and

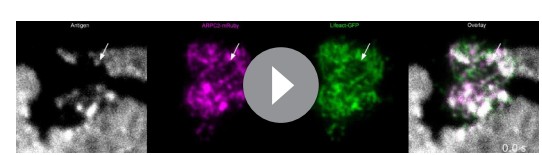

**Video 11.** F-actin and ARPC2 dynamics during uptake of an antigen cluster. The panels show a TIRF microscopy timelapse of a Ramos cells expressing Lifeact-GFP (green) and ARPC2-mRuby (magenta) interacting with a PMS loaded with anti-IgM F(ab')2 (white). Arrow indicates antigen cluster being taken up by the cell.
https://elifesciences.org/articles/48093#video11

**Video 12.** F-actin and DIAPH1 dynamics during uptake of an antigen cluster. The panels show a TIRF microscopy timelapse of a Ramos cells expressing Lifeact-GFP (green) and DIAPH1-mRuby (magenta) interacting with a PMS loaded with anti-IgM F(ab')2 (white). Arrow indicates antigen cluster being taken up by the cell.
https://elifesciences.org/articles/48093#video12

formins. The foci are selectively marked by stoichiometric incorporation of ARPC2 and they disperse after inhibition of Arp2/3 function, consistent with a branched-actin structure. The importance of the Arp2/3 complex in foci formation is consistent with previous data linking activators of the Arp2/3 complex, WAS WIPF1 and ITSN2, with foci formation in T cells (*Janssen et al., 2016*; *Kumari et al., 2015*) and in B cells (*Burbage et al., 2018*; *Keppler et al., 2015*). In contrast, formins, such as DIAPH1, localize less specifically to the foci and their activity instead generates interspersed actin fibers. Thus, both Arp2/3 and formins are active throughout the B cell synapse during antigen uptake without large-scale segregation. Our data thus extend previous knowledge of the importance of Arp2/3 complex in the generation of the lamellipodial actin ring and centripetal membrane flow important for B cell adhesion, BCR signaling and B cell activation (*Bolger-Munro et al., 2019*).

Our data also show that Arp2/3-generated actin foci are intimately associated with antigen extraction, as seen in instances of their presence at the sites of antigen endocytosis and their close, albeit stochastic, relationship to BCR tyrosine-phosphorylated proteins and CCPs. Our model thus suggests that forces that lead to antigen extraction are dependent on new actin polymerization in actin foci. This hypothesis is supported by previous observation that F-actin is continuously accumulating on individual membrane invaginations during B cell antigen uptake (*Natkanski et al., 2013*). If the plasma membrane is instead pulled inward by myosin IIa contractility, F-actin should accumulate before the membrane starts invaginating and then move inward during invagination. In addition, our 3D superresolution imaging shows that actin is associated with the sides of the antigen clusters during internalization, where polymerization could contribute to membrane invagination. Foci-like structures were also associated with antigen extraction from gel-like substrates in a recent study, where they were found protruding into the substrate (*Kumari et al., 2019*). It is thus possible that Arp2/3 generated actin structures can both push and pull on the antigen presenting cell, eventually using actin polymerization for antigen extraction.

Nevertheless, our data also indicate that the branched actin polymerization in the foci is not sufficient to extract antigen as inhibitions of formin and myosin IIa activities also inhibit antigen uptake. Since formin inhibition altered foci dynamics, we hypothesize that formins together with myosin IIa provide stability to actin polymerization in the foci that helps the branched actin to expand in between the future base and tip of the invagination, generating an inward force, as opposed to a lateral force and a comet-like movement along the plasma membrane. A similar stabilizing role for myosin contractility was suggested for function of CCPs in other cells types (*Chandrasekar et al., 2014*). Alternatively, myosin contractility or fiber dynamics may help to dislodge the antigen from the presenting membrane before endocytosis through actin foci. Finally, it is possible that formin-mediated actin polymerization out of the foci extends the range of movement of the foci during late stages of antigen endocytosis. The latter two points are supported by the observations of cells after pharmacological inhibition of the Arp2/3 complex, when the residual fibers are able to transport antigen clusters and pull large membrane ridges or tubules from the presenting membrane both by their contractile motion and by their growth. Future studies will be needed to dissect the role of these mechanisms further and also to determine the exact roles of specific formin family members for these processes as our knock outs of DIAPH1 and FMNL1 were not sufficient to fully recapitulate formin inhibitor treatment.

The actin organization and mechanism of antigen extraction studied here pertains primarily to the synapses formed by naive and memory B cells, where antigen extraction is directly followed by endocytosis (*Nowosad et al., 2016*). Another mechanism to extract antigen, not directly coupled to endocytosis, exists in germinal center B cells. In these cells, peripheral actomyosin-rich pod like protrusions extract antigen and transport it by outward actin flow along the plasma membrane away from the antigen-presenting surface (*Kwak et al., 2018*; *Nowosad et al., 2016*). The regulation of actin in these structures has not yet been investigated, but may also involve Arp2/3-generated

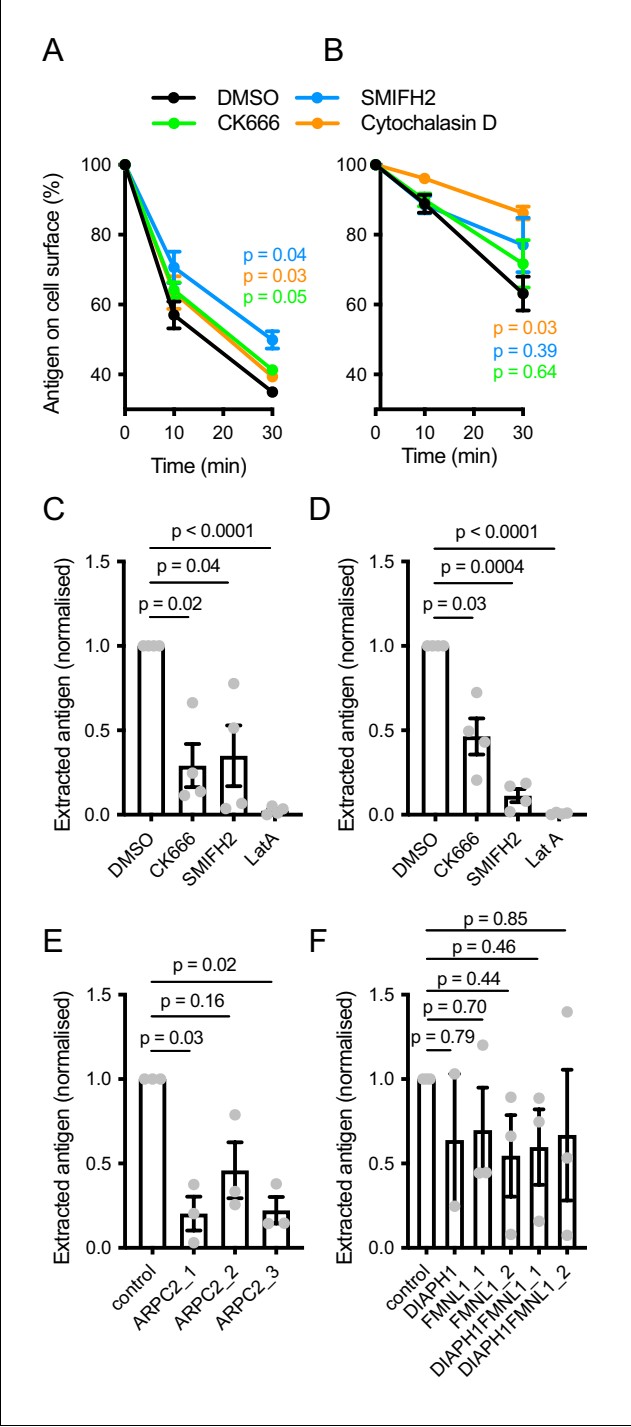

**Figure 5.** Importance of actin polymerization pathways in membrane antigen extraction. (**A, B**) Internalization of soluble anti-Igκ F(ab')$_2$ by naive mouse B cells (**A**) or soluble anti-IgM F(ab')$_2$ by Ramos cells treated with the indicated inhibitors. Data are means ± SEM of n = 3 (**A**) or 4 (**B**) experiments. P, significance of the 30 min timepoint in two-way ANOVA comparing treated cells to DMSO. (**C, D**) Extraction of PMS-bound anti-Igκ F(ab')$_2$ by naive mouse B cells (**C**) or anti-IgM F(ab')$_2$ by Ramos cells (**D**) treated with inhibitors. Data points represent mean cell values from individual experiments normalized to DMSO controls. Bars show means ± SEM n = 4 experiments. (**E, F**) Extraction of PMS-bound anti-IgM F(ab')$_2$ by Ramos cells targeted with the indicated gRNAs. Data points represent mean cell values from individual experiments normalized to controls. Bars show means ± SEM n = 2–3 experiments. In C-F, p values from one-way repeated measures ANOVA are shown. The online version of this article includes the following figure supplement(s) for figure 5:

*Figure 5 continued on next page*

*Figure 5 continued*

**Figure supplement 1.** Internalization of soluble transferrin (Tf) by naive mouse B cells in the presence of the indicated inhibitors.

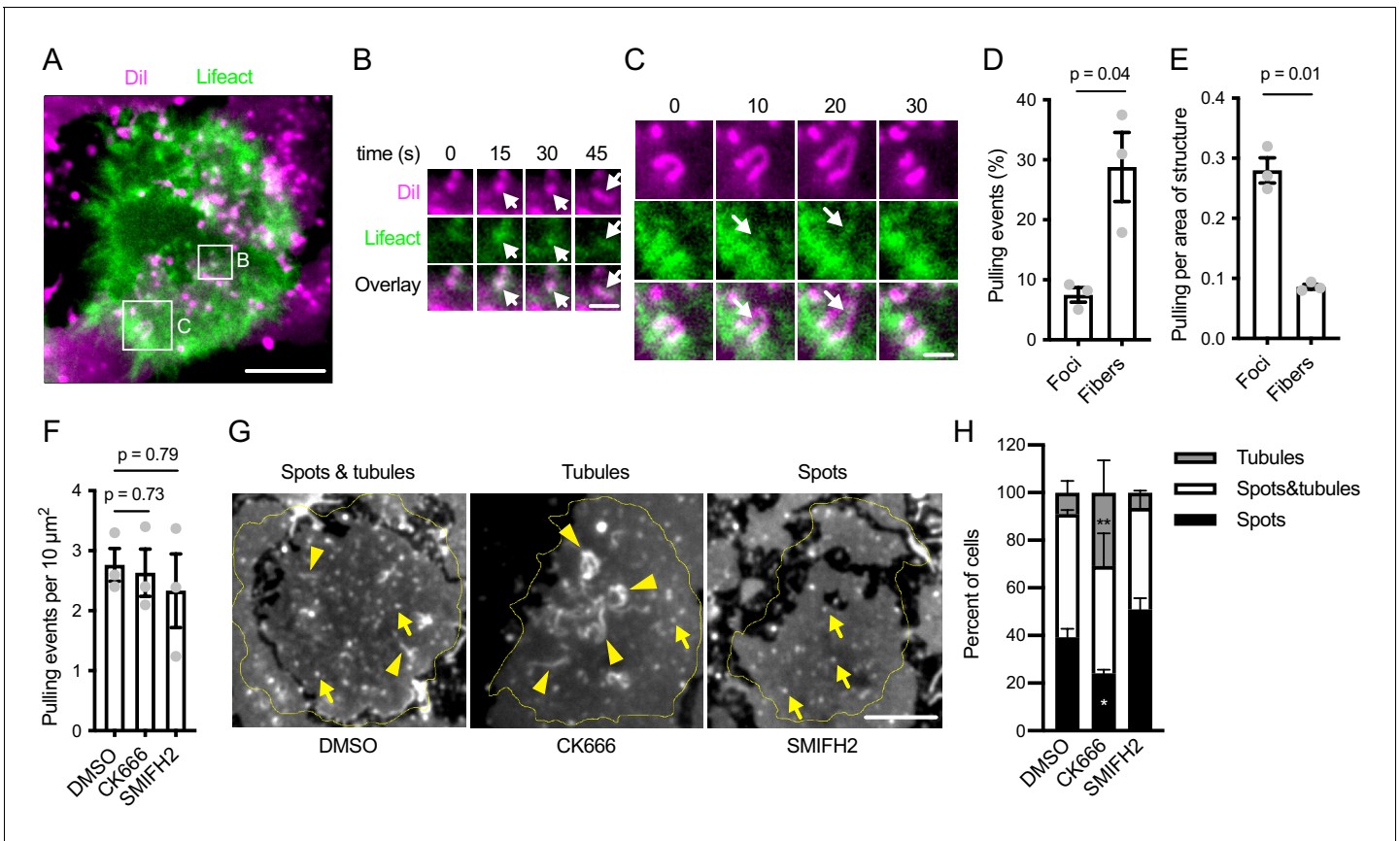

**Figure 6.** Effect of actin structures on mechanical activity during antigen extraction. (A) TIRF image of a Lifeact-GFP-expressing Ramos cell (green) interacting with anti-IgM F(ab')2-loaded PMS stained with DiI (magenta). Spots and tubules in the DiI image indicate deformations of the PMS by the cell. A region containing a DiI spot associated with an actin focus (arrow), later elongating into a fiber is magnified in (B), another region in which a fiber (arrow) elongates a tubule is shown in (C). In B, C the channel intensities were rescaled to the local ranges. Time is shown as relative duration from the initial frame shown. (D) Quantification of the percent of pulling events associated with actin foci or fibers. (E) The rate of pulling events normalized to the area of the cells covered by the two actin structures in the same experiments as in E. Data points in D, E are means of 28–40 cell values from individual experiments, bars show means ± SEM of n = 3 experiments. P, significance in paired t-tests. (F) Quantification of the total number of pulling events in Ramos cells treated with DMSO or the indicated inhibitors. Data points are means of 28–40 cell values from individual experiments, bars show means ± SEM of n = 3 experiments. P, significance in one-way ANOVA. (G) Examples of the shape of membrane deformations in Ramos synapses used to quantify the effect of the inhibitors. Yellow lines show cell outlines. Arrows indicate examples of DiI spots, arrowheads examples of tubules. (H) Quantification of the frequency of synapses containing the indicated membrane shapes in Ramos cells treated with DMSO or the inhibitors. Data are means ± SEM from n = 3 experiments. *, p=0.03, **, p=0.003 in two-way ANOVA against the DMSO control. Scale bars, 5 μm (A, G), 1 μm (B, C).

The online version of this article includes the following video(s) for figure 6:

**Figure 6—video 1.** Relationship of mechanical activity and F-actin in the B cell synapse.
https://elifesciences.org/articles/48093#fig6video1

**Figure 6—video 2.** Mechanical activity in a DMSO-treated Lifeact-expressing Ramos cell on an anti-IgM F(ab')2-loaded PMS labeled with DiI.
https://elifesciences.org/articles/48093#fig6video2

**Figure 6—video 3.** Mechanical activity in a CK666-treated Lifeact-expressing Ramos cell on an anti-IgM F(ab')2-loaded PMS labeled with DiI.
https://elifesciences.org/articles/48093#fig6video3

**Figure 6—video 4.** Mechanical activity in a SMIFH2-treated Lifeact-expressing Ramos cell on an anti-IgM F(ab')2-loaded PMS labeled with DiI.
https://elifesciences.org/articles/48093#fig6video4

branched actin as they resemble miniature lamellipodia. Arp2/3-dependent actin polymerization is therefore likely a general requirement for antigen extraction from antigen-presenting cells.

Unexpectedly, our data show that the formation of the actin foci in naive B cells does not require antigen stimulation. The actin foci may thus be generated continuously as a general property of the B cell cortical cytoskeleton. In agreement with this observation, the majority of actin foci in the synapse do not associate with antigen clusters, or with CCPs. However, a small fraction of actin foci did associate with antigen clusters and to a lower degree with CCPs. Since BCR signaling regulates actin dynamics (*Tolar, 2017*), including foci dynamics shown here, as well as CCP dynamics (*Natkanski et al., 2013*; *Stoddart et al., 2002*) we speculate that a local synergy between BCR signaling, actin dynamics and CCP formation at BCR clusters either enhances the rates of formation of endocytic structures, or stabilizes otherwise abortive events to promote efficient antigen uptake. More detailed observations of BCR signaling during interaction of BCR clusters with foci and CCPs will be needed in the future.

Our data establish a new view on the architecture of B cell synapses and on the molecular basis driving its formation. This information can be used to better understand the role of F-actin not just in B cell antigen endocytosis, but also in receptor signaling at the plasma membrane. The organization of the BCR, and other signaling receptors on B cell surfaces critically depend on actin structure (*Mattila et al., 2016*) and defects in actin polymerization contribute to the dysregulation of signaling in primary immunodeficiencies and in autoimmunity (*Tolar, 2017*). Our studies should guide future investigations of the regulation Arp2/3 and formin pathways in B cells in this context.

## Materials and methods

### Mice

C57BL/6 mice and heterozygous Lifeact-EGFP mice on a C57BL/6 background were used as a sources of primary mouse B cells. All mice were between 1–6 months old and were male and female. Mice were bred and treated in accordance with guidelines set by the UK Home Office (project license number 7008844) and the Francis Crick Institute Ethical Review Panel.

### B cell purification

Primary mouse B cells were isolated from spleens by mashing tissue through a 70 µm cell strainer, followed by red blood cell lysis using ACK Lysing Buffer (Gibco), and negative selection using anti-CD43 microbeads (Miltenyi Biotec).

### Cell lines

HEK293T and Ramos cell lines were provided, authenticated and mycoplasma-tested by the Francis Crick Institute Cell Services.

### Surrogate antigens

Anti-BCR antibodies used to stimulate B cells on PMSs and in solution were goat anti-mouse Igκ F(ab')$_2$(Southern Biotech) for mouse splenic B cells and goat F(ab')$_2$ anti-humanFc5µ (Jackson Immunoresearch) for Ramos cells. The antibodies were biotinylated with 20-fold molar excess of NHS-LC-LC-biotin (Pierce) and labeled with either Cy3 or Cy5 Monoreactive dyes (GE Healthcare), or with AlexaFluor 405 or AlexaFluor 647 NHS esters (ThermoFisher) in Sodium Carbonate buffer. Excess dye was removed using Zeba 7K MWCO desalting columns (Pierce, ThermoFisher).

### Antibodies

All antibodies used in this study were:

| Antibody | Clone | Source | Final concentration | Use |
|---|---|---|---|---|
| Alexa Fluor 488 Mouse anti-BLNK | J117-1278 | BD Biosciences | 1:10 | IF |

*Continued on next page*

*Continued*

| Antibody | Clone | Source | Final concentration | Use |
|---|---|---|---|---|
| Alexa Fluor 488 Phalloidin | Catalogue number: A12379 | Invitrogen | 1:250 | IF |
| Alexa Fluor 568 Phalloidin | Catalogue number: A12380 | Invitrogen | 1:250 | IF |
| Alexa Fluor 647 Phalloidin | Catalogue number: A22287 | Invitrogen | 1:250 | IF |
| Anti-rabbit (H+L), IgG F(ab')₂ Fragment (Alexa Fluor 488 Conjugate) | (Secondary antibody) | Cell Signaling | 1:2000 | IF |
| Anti-rabbit (H+L), IgG F(ab')₂ Fragment (Alexa Fluor 647 Conjugate) | (Secondary antibody) | Cell Signaling | 1:2000 | IF |
| ARPC2 antibody | Polyclonal | GeneTex | 1:1000 | IB |
| Brilliant Violet 421 anti-human CD19 Antibody | HIB19 | BioLegend | 1:200 | IF |
| DIAPH1 antibody | Polyclonal | GeneTex | 1:3000 | IB |
| FITC Rat Anti-Mouse CD45R/B220 | RA3-6B2 | BD Biosciences | 1:200 | IF |
| FMNL1 antibody | [C1C2], Internal, Polyclonal | GeneTex | 1:3000 | IB |
| GAPDH Antibody | 6C5 | Santa Cruz Biotechnology | 1:1000 | IB |
| Goat anti-Mouse IgG (H+L), Alexa Fluor 680 | (Secondary antibody) | Invitrogen | 1:1000 | IB |
| Goat F(ab')₂ anti-human IgM | Fcm Fragment Specific | Jackson ImmunoResearch | 1 mg/ml | FC, IF |
| Goat F(ab')₂ anti-mouse Igk | (Antigen) | Southern Biotech | 1 mg/ml | FC, IF |
| IRDye 800CW Goat anti-Rabbit IgG (H+L) | (Secondary antibody) | LI-COR | 1:1000 | IB |
| LIVE/DEAD Fixable Violet Dead Cell Stain Kit | Catalogue number: L34958 | Invitrogen | 1:1000 | FC |
| MHC Class I (H-2Kb), Biotin | AF6-88.5.5.3 | eBioscience | 1:1000 | IF |
| MHC Class II (I-A/I-E), Biotin | M5/114.15.2 | eBioscience | 1:1000 | IF |
| Phospho-CD19 Antibody | (Tyr531) | Cell Signaling | 1:50 | IF |
| Phospho-Myosin Light Chain 2 | (T18/S19) | Cell Signaling | 1:50 | IF |
| Phospho-Tyrosine Mouse mAb | (P-Tyr-100) | Cell Signaling | 1:50 | IF |

## Inhibitors

Inhibitors were dissolved in dimethyl sulfoxide (DMSO, Sigma). For all inhibitor experiments, B cells and inhibitors were incubated in HBSS with 0.01% BSA. Appropriate concentrations of DMSO were used for control experiments. The inhibitors and their concentrations were:

| Inhibitor | Target | Source | Final concentration |
| --- | --- | --- | --- |
| Blebbistatin | Myosin II | Caymann Chemical | 50 µM |
| CK666 | ARP2/3 | Merck | 50 µM |
| Latrunculin A | Actin polymerisation | Merck | 2 µM |
| P505-15 | SYK | BioVision | 10 µM |
| Para-Nitroblebbistatin | Myosin II | Optopharma | 50 µM |
| PP2 | SRC-family kinases | Merck | 50 µM |
| SMIFH2 | Formins | Merck | 20 µM |

## Preparation of samples for imaging

Plasma membrane sheets were generated as described (*Nowosad and Tolar, 2017*). In brief, a Lab-Tek 8-well glass-bottomed imaging chambers (Thermo Fisher Scientific) were coated with poly-L-lysine (Sigma), and $2 \times 10^5$ HEK293T cells were added to each well and cultured in full DMEM overnight. The cells were then gently washed with PBS and sonicated with a probe sonicator. Exposed glass surfaces were blocked with 1% BSA in PBS for 1 hr and the chambers were then incubated for 30 min with 24 nM biotinylated annexin V-biotin (BioVision) in Hank's buffered saline solution (HBSS) supplemented with 0.1% bovine serum albumin (BSA). After washing with HBSS-BSA, PMSs were incubated with 1 µg/ml streptavidin for 15 min followed by incubation with surrogate antigens.

Isolated B cells ($1.5 \times 10^6$) or cultured Ramos cells ($2 \times 10^5$) were washed and resuspended in 50 µl of warm HBSS-BSA, added to each well and incubated at 37°C for 20 min to allow immune synapses to form. For live cell imaging, samples were kept at 37°C by a heated chamber and an objective heater. For inhibitor experiments in live cells, inhibitors were added directly to the center of the imaging well at this time. Alternatively, cells were fixed in 2% paraformaldehyde (PFA, Alfa Aesar) for 20 min at room temperature (RT) and washed with HBSS-BSA.

## Live and fixed cell imaging

Epifluorescence and TIRF imaging were carried out either on a Nikon Eclipse Ti microscope with an ORCA-Flash 4.0 V3 digital complementary metal-oxide semiconductor (CMOS) camera (Hamamatsu Photonics) and 100x TIRF objective (Nikon), or on an Olympus IX81 microscope with an Andor iXon electron multiplying charged-coupled device (EMCCD) camera and 100x objective (Olympus). Both microscopes were controlled through Metamorph software (Molecular Devices) and illumination was supplied by 405, 488, 552 and 637 nm lasers (Cairn) through an iLas2 Targeted Laser Illuminator (Gataca Systems) which produces a 360° spinning beam with adjustable TIRF illumination angle. Acquired datasets were analyzed using ImageJ and Matlab (Mathworks). All images were aligned and, where necessary, corrected for background, flatfield illumination and photobleaching.

## Super-resolution imaging

To label F-actin for super-resolution imaging, samples prepared as above blocked for 1 hr using 10% BSA and 5% vol/vol normal mouse serum in HBSS and then incubated with phalloidin-AF647 (1:10 dilution) in HBSS-BSA overnight at 4°C. Just before imaging, cells were washed with PBS and incubated with STORM imaging buffer composed of 6.8 mM Tris (pH 8.0), 1.36 mM NaCl, 1.36% Glucose (wt/vol), 13.4 mM cysteamine (Sigma), 34 mM HCl, 4.76 µM Type II Glucose Oxidase (sigma), 185 nM Catalase from bovine liver (sigma), 2 mM COT (Sigma). STORM imaging was carried out on a Nikon N-STORM microscope (Eclipse Ti-E Inverted Microscope) with an Andor iXon EMCCD camera and 100x/1.49 numerical aperture (NA) oil-immersion TIRF objective. Cells were imaged under TIRF illumination, with a 640 nm laser and 405 nm laser. The 405 nm laser was used to maintain photo-blinking by gradually increasing its power. Fluorescence was collected at

wavelengths between 640 and 790 nm. 10,000 to 20,000 frames were recorded at 18 ms exposure time per frame. For 3D STORM, astigmatic lens in the Nikon N-STORM microscope was slotted into the imaging path and cells were imaged under near-TIRF illumination. 3D calibration was carried out using the Nikon Elements software using fluorescent microspheres adsorbed on a glass coverslip. Molecular coordinates were localized using ThunderSTORM software (*Ovesný et al., 2014*), drift-corrected and displayed using average shifted histograms. 3D STORM localizations were subsequently grouped into 50 nm artificial z-stacks. To produce 3D STORM and epifluorescence image composites, a single epifluorescence image with the appropriate z-coordinate was paired with each 50 nm 3D STORM image. Images were aligned in ImageJ to produce an epifluorescence and 3D STORM composite z-stack. Side view reconstructions were produced from composite z-stacks using maximal intensity projections of a region of a cell along the x or y axis.

## Image analysis of actin foci and fibers

Foci in F-actin images were quantified by a combination of image filtering and tracking in individual cells using Matlab (*Figure 1—figure supplement 1*, see also Image Segmentation *Source code 1*). All cells in all images were automatically identified and segmented based on Lifeact or phalloidin intensity. No cells were excluded from analysis provided they passed the intensity threshold. Foci were enhanced by convolution with a 325 or 450 nm gaussian-shaped filter for primary and Ramos B cells, respectively, and the convoluted images were thresholded by setting pixels below a threshold value, defined as a fraction of the cell's mean actin intensity, to zero. Fibers were enhanced separately by taking the maximum value of images convoluted with 325 or 450 nm line-shaped filters (for primary and Ramos B cells, respectively) rotated at 18-degree increments to account for filament of all orientations. Candidate foci were then detected as areas where the foci mask pixel values exceeded the fiber mask values and further cleaned up by removing small and highly elongated objects. For live cell images, the foci candidates were then tracked through the timelapse and foci not persisting for at least two frames were discarded. Final foci numbers were counted as the mean number of foci per time point and normalized per area of the cell. Foci displacement was calculated as the distance between the start and end positions divided by lifetime. Fibers were identified as areas where the fiber mask exceeded a threshold defined as a fraction of the cell's actin intensity. Pixels belonging to foci were excluded from the fiber areas. The threshold values were adjusted between different experiments carried out on different cells or acquired with different instruments, but were kept constant for all conditions within each experiment.

To compare automated segmentation of actin foci to their manual identification, two primary B cell timelapses were segmented by the automated analysis and by four researchers knowledgeable of the actin structures, but blinded to the results of the automated segmentation (*Figure 1—figure supplement 1B*). We compared the total number of foci detected in the timelapses and also the centers of their positions using a 5-pixel and 1-frame tolerance. 'True positive rate' of the automated segmentation was calculated compared to each human researcher as the percent of the human-detected foci that were also detected by the computer. 'False positive rate' of the automated segmentation was calculated as the percent of foci detected by the computer but not detected by the human. As a comparison, foci positions obtained by the human researchers were also compared to each other using the same calculations. We found that the differences between the computer and the humans were not significantly different from differences between the human researchers themselves. Thus, the automated segmentation provided an unbiased high-throughput foci detection tool with quality similar to manual segmentation.

## Image quantification of enrichment and colocalization

Foci and fibers were first segmented in automatically detected cells in all images as described above. The enrichment of fluorescence signal of interest in actin foci and fibers was calculated from the fluorescent intensity in these structures, normalized to mean intensity in the synapse. Coincidence of actin foci with ARPC2 spots, DIAPH1 spots or CCPs and vice versa was calculated as the fraction the foci that overlapped with the center of the other structures. As a control, centers of the same number and size of structures were placed without overlap at random location in the same cell. A mean of ten of the randomization runs was taken as the random value for each cell.

## Quantification of B cell mechanical activity

Mechanical activity in the synapse was visualized using DiI-labeled PMSs as described (*Natkanski et al., 2013*). Briefly, PMSs were labeled using DiI and imaged in TIRF timelapses for dynamic membrane deformations that appear as local increase in fluorescence intensity. To quantify these membrane pulling events, bright DiI structures were located and tracked using previous algorithms (*Tolar et al., 2009*). To exclude preexisting membrane structures on the PMSs not generated by the B cells, only structures that appeared or moved during the interaction with the cells were included in the analysis. To quantify the total number of pulling events, the number of the DiI structures were normalized per area of the synapse in each frame. To associate pulling events with actin structures, actin foci and fibers were segmented in the same cells and the pulling events were associated with either or both of them based on their center overlapping within the actin structures anytime during their lifetime. To quantify the frequency of synapses containing only DiI spots, a mixture of tubules and spots, or predominantly tubules, synapses were manually classified based on the three examples shown in *Figure 6G* and *Figure 6—video 2–4*.

## Quantification of substrate-bound antigen internalization

For quantification of antigen uptake from PMSs, cells were incubated with inhibitors for 15 min at 37°C before incubation with PMSs in the continued presence of the inhibitors for 30 min. Large-scale image datasets and their processing was performed as described (*Nowosad et al., 2016*). In brief, experimental images were cropped, aligned, background subtracted and corrected for flatfield and spectral bleedthroughs. Cells were segmented using B220 staining (primary B cells) or CD19 (Ramos cells). CellScore (*Nowosad et al., 2016*) was used to gate on live B cells. To detect extracted antigen in the z-stack images, each image was bandpass-filtered and antigen clusters were identified in planes above the synapse by a user- specific global threshold. Total fluorescence of extracted antigen was calculated as the sum of pixel intensities of the background-corrected extracted clusters.

## Soluble antigen and transferrin internalization assay

Isolated B cells or Ramos cells taken from culture were incubated with a LIVE/DEAD marker in PBS for 20 min. They were then washed and incubated with fluorescently labeled, biotinylated surrogate antigens at 1 µg/ml in PBS with 1% BSA and 2 m EDTA for 30 min on ice. Alternatively, the cells were incubated with fluorescently-labeled, biotinylated transferrin (Rockland). Cells were washed and incubated with inhibitors for 20 min on ice. Samples were then split into three and incubated either on ice or at 37°C for 10 or 30 min before fixation with 2% PFA on ice for 20 min. After fixation, cells were washed and stained for 20 min on ice with fluorescently labeled streptavidin, to label the ligands remaining on the cell surface. Samples were analyzed on an LSR Fortessa (BD Biosciences). Internalization was quantified as the percentage of antigens remaining at the cell surfaces using FlowJo (TreeStar).

## Plasmids, cloning, transfections and CRISPR/Cas9 gene targeting

Human ARPC2-mRuby3 was constructed from pLVX-ARPC2-GFP and pOPINF2-WIPI2B-human-mRuby3 and cloned into pLVX for lentivirus-mediated transfection. To construct mRuby-DIAPH1, we cloned human DIAPH1 cDNA from Caco-2 cells using RNAeasy micro and QuantiTect Reverse Transcription kits (Quiagen) and inserted it into pVLX-mRuby3. For Lifeact-GFP transfection we used pLVX-LifeactGFP plasmid with the resistance gene switched to neomycin. Clathrin light chain A-mCherry was constructed from mouse Clta-GFP (*Natkanski et al., 2013*) and inserted into pLenti-puro. CRISPR sgRNA sequences were designed using the Broad Institute's sgRNA Designer. Forward and reverse oligonucleotides including the guide sequence were synthesised, phosphorylated, annealed and individually cloned into lentiGuide-Puro or lentiGuide-Neo plasmids.

Guide sequences were:

| Guide | Forward oligonucleotide | Reverse oligonucleotide |
| --- | --- | --- |
| ARPC2_1 | CACCGACAATGGAATCCTTGGATGC | AAACGCATCCAAGGATTCCATTGTC |
| ARPC2_2 | CACCGTTTCCTCACAGATTTCGATG | AAACCATCGAAATCTGTGAGGAAAC |

*Continued on next page*

*Continued*

| Guide | Forward oligonucleotide | Reverse oligonucleotide |
| --- | --- | --- |
| ARPC2_3 | CACCGCATTGGAAAGGTGTTCATGC | AAACGCATGAACACCTTTCCAATGC |
| DIAPH1 | CACCGGAGGCATACCCATTCC | AAACGGAATGGGTATGCCTCC |
| FMNL1_1 | CACCGGGGACCACCATGGGCAACG | AAACCGTTGCCCATGGTGGTCCCC |
| FMNL1_2 | CACCGGGTCGCACTCACCAGGGCG | AAACCGCCCTGGTGAGTGCGACCC |

Recombinant replication-incompetent lentiviruses were produced by co-transfecting HEK293T cells using transIT-LT1 (Mirus)-mediated pMD2.G and psPAX2 helper plasmids together with the appropriate lentiviral plasmid. Lentivirus-containing media were harvested 48 and 72 hr following transfection and used to spin-infect Ramos cell lines for 90 min at 1350 x g in the presence of 5 μg/ml polybrene (Sigma). Sixty hours after the spinfection, cells were selected using 2.5 μg/ml of puromycin or 500 μg/ml of geneticin for one to two weeks. Cells expressing the fluorescent markers were further purified using Avalon cell sorter (Propel Labs).

## Immunoblots

Ramos cells were lysed with RIPA buffer (Sigma), containing 1% protease inhibitor cocktail (Sigma), for 10 min on ice and then centrifuged at 13,000 rpm for 10 min at 4°C. Samples were heated in 1X NuPAGE LDS Sample Buffer (Invitrogen) and 1X NuPAGE Sample Reducing Buffer (Invitrogen) at 95C for 5 min and then separated using ExpressPlus PAGE Gels 4–20% (GenScript). Proteins were transferred to a polyvinylidene fluoride (PVDF) membrane and then blocked for 1 hr in 50% Odyssey Blocking Buffer (LI-COR) and 50% PBS. The membrane was incubated with the appropriate concentration of primary antibodies in PBST overnight on a rocker. The membrane was washed with PBST and incubated with the fluorescently conjugated secondary antibody in 50% PBST, 50% Odyssey Blocking Buffer and 0.05% sodium dodecyl sulfate (SDS) for 1 hr. After washing with PBST, the membrane was incubated in PBS and imaged using an Odyssey CLx scanner (LI-COR).

## Acknowledgements

This work was supported by the European Research Council (Consolidator Grant 648228) and the Francis Crick Institute, which receives its core funding from Cancer Research UK (FC001185, FC001209), the UK Medical Research Council (FC001185, FC001209), and the Wellcome Trust (FC001185, FC001209).

## Additional information

### Funding

| Funder | Grant reference number | Author |
| --- | --- | --- |
| Francis Crick Institute | FC001185 | Laabiah Wasim<br>Dessislava Malinova<br>Pavel Tolar |
| H2020 European Research Council | 648228 | Dessislava Malinova<br>Pavel Tolar |
| Francis Crick Institute | FC001209 | Michael Way |

The funders had no role in study design, data collection and interpretation, or the decision to submit the work for publication.

### Author contributions

Sophie I Roper, Conceptualization, Investigation, Methodology; Laabiah Wasim, Dessislava Malinova, Investigation; Michael Way, Methodology; Susan Cox, Supervision, Methodology; Pavel Tolar, Conceptualization, Supervision, Investigation, Methodology

## Author ORCIDs

Sophie I Roper [ID] https://orcid.org/0000-0001-7776-7670
Michael Way [ID] http://orcid.org/0000-0001-7207-2722
Pavel Tolar [ID] https://orcid.org/0000-0003-4693-7299

## Ethics

Animal experimentation: Mice were bred and treated in accordance with guidelines set by the UK Home Office (project license number 7008844) and the Francis Crick Institute Ethical Review Panel.

## Decision letter and Author response

Decision letter https://doi.org/10.7554/eLife.48093.sa1
Author response https://doi.org/10.7554/eLife.48093.sa2

## Additional files

### Supplementary files

- Source code 1. Image Segmentation.

- Transparent reporting form

### Data availability

All data generated or analysed during this study are included in the manuscript and supporting files.

The following previously published datasets were used:

| Author(s) | Year | Dataset title | Dataset URL | Database and Identifier |
|---|---|---|---|---|
| Brazão TF, Johnson JS, Müller J, Heger A, Ponting CP, Tybulewicz VL | 2016 | Long non-coding RNAs in B cells | https://www.ncbi.nlm.nih.gov/geo/query/acc.cgi?acc=GSE72019 | NCBI Gene Expression Omnibus, GSE72019 |
| Klijn C, Durinck S, Stawiski EW, Haverty PM, Jiang Z, Liu H, Degenhardt J, Mayba O, Gnad O, Liu J, Pau G, Reeder J, Cao y, Mukhyala K, Selvaraj SK, Yu M, Zynda GJ, Brauer MJ, Wu TD, Gentleman RC, Manning G, Yauch RL, Bourgon R, Stokoe D, Modrusan Z, Neve RM, Sauvage FJ, Settleman J, Seshagiri S, Zhang Z | 2015 | A comprehensive transcriptional portrait of human cancer cell lines | https://www.ebi.ac.uk/ega/studies/EGAS00001000610 | European Genome-phenome Archive, EGAS00001000610 |

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
