## [Decision Letter]

**Acceptance summary:**

Your work greatly advances the working model for how the actin cytoskeleton is organised to enable antigen extraction by B cells, which is critical to their function in antigen presentation. You have shown how the Arp2/3 branched actin and formin linear actin nucleation factors and myosin cooperate to generate a network of transient foci that overlie the extraction sites and are interconnected by the linear filaments that also incorporate myosin II based contractile units. While important questions remain, this work sets a high standard for quantitative analysis of these dynamic structures generating a strong foundation for future exploration.

**Decision letter after peer review:**

Thank you for submitting your article "B cells extract antigens using a dynamic network of Arp2/3-generated actin foci interconnected with linear filaments" for consideration by *eLife*. Your article has been reviewed by three peer reviewers, and the evaluation has been overseen by a Reviewing Editor and Anna Akhmanova as the Senior Editor. The reviewers have opted to remain anonymous.

The reviewers have consulted with one another and the Reviewing Editor has drafted this decision to help you prepare a revised submission.

Summary:

Your manuscript "B cells extract antigens using a dynamic network of Arp2/3- generated actin foci interconnected with linear filaments" presents an analysis of actin structures associated with BCR signaling and early phases of antigen extraction by B cells on model antigen presenting surfaces. You use cutting edge imaging methods to show suggestive results that BCR signaling is associated with distinct actin structures (compared to the usual lamella and lamellipodia). Subsequently you assemble circumstantial evidence using imaging and CRISPR generated knockdowns that these structures may be implicated in antigen extraction. However, a number of key experiments and critical controls are required to establish your novel claims and the reviewers have consulted to suggest experiments that can be accomplished in 2 months.

Essential revisions:

1) Using foci and linear actin tracking in live cells, you should show that the Arp2/3-dependent foci and the Formin- dependent filaments are part of a crosslinked network. It will be crucial to include a membrane marker in this experiment to discern filopodia from formin-based linear filament population.

2) Using your previously published tension sensors, and Arp2/3 inhibition/knockdown, the authors should measure forces exerted by the foci during antigen endocytosis. Knowing the magnitude of the forces at the antigen site would bridge the missing connection between the localization of foci and antigen endocytosis.

3) By examining the endocytosis of a generic clathrin pathway marker such a transferrin, you need to substantiate the specific impact of Arp2/3 inhibition on antigen internalization.

4) You should provide a supplementary figure detailing the workflow utilized to extract foci, and how the Gaussian filter size for enhancing foci was determined/ chosen. The efficiency of this foci extraction method also needs to be shown.

[Editors' note: further revisions were requested prior to acceptance, as described below.]

Thank you for resubmitting your work entitled "B cells extract antigens using Arp2/3-generated actin foci interspersed with linear filaments" for further consideration by *eLife*. Your revised article has been evaluated by Anna Akhmanova as the Senior Editor, a Reviewing Editor and three peer reviewers.

The manuscript has been improved but there are some remaining issues that need to be addressed before acceptance, as outlined below:

Please see the comments below. After consultation, we think that an excellent title would be "B cells extract antigens at Arp2/3-generated actin foci interspersed with linear filaments" but you can suggest other titles that accommodate reviewer 2 concerns.

Reviewer #1:

The manuscript has substantially improved after the revision and provides new insights into antigen extraction at B cell synapse. I have no further experimental suggestions for the authors, and strongly recommend the manuscript for publication at *eLife*.

Reviewer #2:

The authors have addressed many of the major comments from the previous review. Their improved description of the determination of foci and fibers, inclusion of controls, and analysis of PMS deformation to infer the location and directionality of forces make the manuscript more complete. The characterization of this connected network in B cells is novel and suggestive of a role for this network in B cell signaling.

However, the significance of the observed phenomena and its role in B cell function is still not clear to this reviewer. The experiments in the original and the additional experiments in revision seem to imply that these actin foci and filaments are a general feature of B cell cytoskeleton and only mildly dependent on BCR signaling.

So the statement: "We propose that when B cells extract antigens from live antigen-presenting cells, actin foci and fibers form nodes and edges in a loosely connected network, whose function is required for efficient antigen extraction. " is speculative as best.

As stated by the authors, these structures do not assemble in an obligatory manner either at antigen clusters or at CCPs, which are the sites of endocytosis. Therefore, their relevance for Ag gathering is not apparent. The data in Figure 5 suggests that the loss of Arp23 or mDia function leads to generic deficits in Ag gathering, which are consistent with general defects in endocytosis as shown in control Tf data. From all of these it is difficult to conclude that these foci play any significant role in exerting forces to test Ag quality or lead to efficient Ag extraction.

Given these concerns, the authors have not convincingly demonstrated that "B cells extract antigens" using these actin structures, and therefore the authors should revise the title to reflect their findings more accurately.

However, the identification of these structures and quantification of their origins and dynamics will be of sufficient interest to the community.

Reviewer #3:

To our opinion, the authors successfully answered most of the requests of the editor. However, we fill that the two following points should be addressed prior to final acceptance.

1) The authors use the DIL membrane marker to observe the PMS invaginations. In Figure 6A they show a figure where DIL patches are visible even outside of the B cell area. How do the authors explain this? Are DIL spots within the synapse more important in number and intensity than the ones outside? This should be clarified.

2) An article describing actin patches at the B cell immune synapse has been published during the time the present article was revised. Although the two articles are complementary, the authors should imperatively cite it and eventually discuss it (Kumari, Pineau et al., 2019).

---

## [Author Response]

Essential revisions:1) Using foci and linear actin tracking in live cells, you should show that the Arp2/3-dependent foci and the Formin- dependent filaments are part of a crosslinked network. It will be crucial to include a membrane marker in this experiment to discern filopodia from formin-based linear filament population.

Our data mainly focused on the functional synergy between then Arp2/3 and formin mediated actin polymerization. To substantiate the physical connections between the actin foci and actin fibers within the synapse, we extended our analysis of Ramos cells co-expressing Lifeact-GFP and ARPC2-mRuby. We now show in a new part of Figure 3 that ARPC2-containing actin foci emanate fibers that often transiently connect to other actin structures including other foci (Figure 3C). We think that this finding illustrates the closely interconnected nature of the actin network in the synapse. This result is also consistent with the detectable presence of DIAPH1 in actin foci (Figure 3D, E, G, I), and with the reduction of interspersed fibers by the formin inhibitor SMIFH2 (Figure 2A). Nevertheless, given the transient nature of the network-like connections, we agree that it is more appropriate to think of the foci-fiber relationship as a mixed pattern with a cooperative function, rather than a “crosslinked network”. We therefore refer to this architecture as a “functional network” throughout the manuscript and toned down the statements on the importance of the physical interconnection of the foci by linear filaments.

To address the possibility that the fibers in the synapse represent filopodia, rather than filaments within the cell, we labeled the plasma membrane of the Lifeact-GFP/ARPC2-mRuby co-expressing Ramos cells with DiD and imaged their immune synapses. The results are summarized in a new Figure 3—figure supplement 1. The DiD showed dynamic undulations of the plasma membrane in the synapse in pattern that partly corresponded to the Lifeact signal. This is consistent with the actin-dependent mechanical activity of plasma membrane of the cells. Filopodia at the cell edges were also visible in these experiments as thin, Lifeact-positive, DiD-positive protrusions. However, despite of the general correspondence of the Lifeact and DiD signals in the synapse, we could find many Lifeact-positive, ARPC2-negative filaments in the synapse that did not accumulate any extra DiD signal. An example is shown in the figure. We conclude that most of the filament dynamics relevant for this paper and for antigen extraction occurs within the cortical cytoskeleton rather than within filopodia.

2) Using your previously published tension sensors, and Arp2/3 inhibition/knockdown, the authors should measure forces exerted by the foci during antigen endocytosis. Knowing the magnitude of the forces at the antigen site would bridge the missing connection between the localization of foci and antigen endocytosis.

We agree that the relationship of the actin architecture to the location of the pulling forces is important. We decided not to use the DNA sensors, however, as their force signal on PMSs is weak because the tension is buffered by movement of the PMS during pulling. Instead, we used our previous technique of directly analyzing the force-mediated PMS deformation using labeling with the lipid dye DiI (Natkanski et al., 2013). As cells pull on the PMS, DiI fluoresce locally increases as the deformed PMS accumulates at the sites of pulling. The DiI fluorescence appears either as spots, which we previously showed are vertically oriented membrane invaginations, or as ridges or tubules, depending on the direction of the pulling and range of the movement. Although the strength of the forces cannot be determined by this analysis, it allows us to robustly detect the sites of the pulling forces, their direction and also the overall vigor of the mechanical activity. We used Lifeact-expressing Ramos cells for these experiments to improve the spatial resolution due to the bigger size of the cells, but pilot experiments suggest that the results translate to primary cells as well. The data are presented in the new Figure 6, in Videos 13-16, and in the corresponding sections of the Results and Materials and methods.

We found that the sites of pulling associated with both actin foci and actin fibers (Figure 6 A, B, C, Video 13). Because fibers cover greater area than foci in the synapse, most of the pulling activity associated with fibers (Figure 6D). However, correcting for the difference in area, foci were the more efficient structure at pulling (Figure 6E). Although both actin structures pulled on the PMSs, there was a difference in the shape of the DiI structures indicating difference in the direction of pulling. While foci were typically associated with DiI spots, which grew inwards, fibers often produced elongated structures, tubules and ridges (Figure 6B, C), that moved laterally. This explains why foci were found to be associated with antigen movement into the cell, but fibers were not.

These results were corroborated by inhibitor treatment. We found that the overall pulling activity was not significantly reduced by either CK666 or SMIFH2 (Figure 6F). Since these drugs did inhibit antigen extraction, these results indicate that mechanical activity per se is not sufficient for antigen internalization. In addition, treatment of the Ramos cells with CK666 led to enhanced generation of tubules/ridges at the expense of spots, with a significant fraction of cells producing excessive tubulation within the synapse (Figure G, H, Video 15). This further confirms that fibers, which are enhanced in CK666-treated cells, are not sufficient for antigen internalization. Possibly, because of their orientation along the plasma membrane, the fibers generate lateral, instead of inward-oriented movement. Because of the length of the tubules generated by fibers, we speculate that the fibers are producing longer range of membrane movement than foci. Although the DiI imaging did not reveal this clearly, we believe that extension of the inward antigen movement by fibers associated with foci contributes to antigen extraction and is part of the reason why formin inhibition with SMIFH2 also blocks antigen uptake. Overall, we believe that these data add important information to the manuscript, namely that while either Arp2/3 or formin activity apply forces on the antigen, their concerted action is needed for pulling to proceed to antigen internalization.

3) By examining the endocytosis of a generic clathrin pathway marker such a transferrin, you need to substantiate the specific impact of Arp2/3 inhibition on antigen internalization.

To address the effects of the inhibitors of on general clathrin-mediated endocytosis, we measured internalization of soluble transferrin. We show in a new Figure 5—figure supplement 1 that the inhibitors had a very modest effect, similar to the modest effect on internalization of the soluble surrogate antigen. We conclude that most of the effects that these drugs exert on the uptake of membrane antigen is specific to the immune synapse due to increased demands on the actin cytoskeleton.

4) You should provide a supplementary figure detailing the workflow utilized to extract foci, and how the Gaussian filter size for enhancing foci was determined/ chosen. The efficiency of this foci extraction method also needs to be shown.

To illustrate the image segmentation more clearly, we created a new figure (Figure 1—figure supplement 1A), which shows the workflow of the algorithm, the shape and size of the filtering elements, and the intermediate results that are used to compute the final segmentation masks identifying foci and fibers. In addition to explanation in the figure legend, we also extended the description of the technique in the Materials and methods and we include the Matlab source code used for the segmentation and for foci counting.

We agree that understanding of the performance of the segmentation is important to interpret the results. We therefore compared the automated segmentation with manual segmentation by four human researchers who were knowledgeable of the actin structure (they were co-authors of the manuscript), but had no prior knowledge of the results of the computerized segmentation. We focused on foci segmentation in two Lifeact-GFP timelapses of primary B cells as fibers were found to be too laborious to segment manually even for such a limited number of images. We compared the results of the segmentation by the computer to that of the humans and also the humans to each other. The results are shown in Figure 1—figure supplement 1B. We found that computerized segmentation detected more foci than manual segmentation, indicating that it was more sensitive. To evaluate the positions of the detected foci, we calculated “true positive rates” of the computerized segmentation as the percent of manually segmented foci that were also detected by the computer, and “false positive rates” as the percent of foci detected by the computer, but not detected manually. We calculated these rates for each computer-human pair and also for all human-human pairs. The results showed that the computer-human pairs could not be distinguished in their true and false positive rates from the human-human pairs. Thus, although not perfect, the automated segmentation performs just as well as any experienced researcher, while being consistent and higher throughput. It is also more sensitive, which is likely because human researchers are quickly fatigued by the process after a few image frames.

[Editors' note: further revisions were requested prior to acceptance, as described below.]

The manuscript has been improved but there are some remaining issues that need to be addressed before acceptance, as outlined below:Please see the comments below. After consultation, we think that an excellent title would be "B cells extract antigens at Arp2/3-generated actin foci interspersed with linear filaments" but you can suggest other titles that accommodate reviewer 2 concerns.Reviewer #2:[…] However, the significance of the observed phenomena and its role in B cell function is still not clear to this reviewer. The experiments in the original and the additional experiments in revision seem to imply that these actin foci and filaments are a general feature of B cell cytoskeleton and only mildly dependent on BCR signaling.So the statement: "We propose that when B cells extract antigens from live antigen-presenting cells, actin foci and fibers form nodes and edges in a loosely connected network, whose function is required for efficient antigen extraction. " is speculative as best.As stated by the authors, these structures do not assemble in an obligatory manner either at antigen clusters or at CCPs, which are the sites of endocytosis. Therefore, their relevance for Ag gathering is not apparent. The data in Figure 5 suggests that the loss of Arp23 or mDia function leads to generic deficits in Ag gathering, which are consistent with general defects in endocytosis as shown in control Tf data. From all of these it is difficult to conclude that these foci play any significant role in exerting forces to test Ag quality or lead to efficient Ag extraction.Given these concerns, the authors have not convincingly demonstrated that "B cells extract antigens" using these actin structures, and therefore the authors should revise the title to reflect their findings more accurately.However, the identification of these structures and quantification of their origins and dynamics will be of sufficient interest to the community.

We agree that the relationship of the actin structures to antigen extraction is complex, however, we think that we show sufficient evidence that actin foci are sites of antigen endocytosis, despite their stochastic appearance. We have therefore changed the title to the one suggested by the editor to better reflect our conclusions: “B cells extract antigens at Arp2/3-generated actin foci interspersed with linear filaments”.

To acknowledge the speculative nature of the statement on line 356, we edited the sentence to start with “We speculate that …”

Reviewer #3:To our opinion, the authors successfully answered most of the requests of the editor. However, we fill that the two following points should be addressed prior to final acceptance.1) The authors use the DIL membrane marker to observe the PMS invaginations. In Figure 6A they show a figure where DIL patches are visible even outside of the B cell area. How do the authors explain this? Are DIL spots within the synapse more important in number and intensity than the ones outside? This should be clarified.

Indeed, DiI spots were observed on PMSs outside of B cell synapses. These are spots present on the PMSs before interaction with B cells. We believe that these spots are pre-existing ruffles, bulges or tethered vesicles of the PMSs. For this reason, to analyze the mechanical activity within B cell synapses we used live cell imaging and tracked only DiI spots that newly appeared or moved during the recording time. Thus, the results in Figure 6 reflect only the pulling activity of the B cells and are not affected by the preexisting DiI spots on the PMSs unless the B cells pull on them as well. This is explained in “Materials and methods” under “Quantification of B cell mechanical activity”. Please note that some of the cells lost LifeAct-GFP expression and thus they are not visible, yet still pull on certain areas of the PMSs outside of the analyzed synapses.

2) An article describing actin patches at the B cell immune synapse has been published during the time the present article was revised. Although the two articles are complementary, the authors should imperatively cite it and eventually discuss it (Kumari, Pineau et al., 2019).

We have added a citation of the paper by Kumari et al. along with a brief discussion at the end of the fourth paragraph of “Discussion”.